# Cell-type-specific co-expression inference from single cell RNA-sequencing data

Chang Su [1,3], Zichun Xu [1,5], Xinning Shan [1], Biao Cai [1,4], Hongyu Zhao [1] ✉ & Jingfei Zhang [2] ✉

The advancement of single cell RNA-sequencing (scRNA-seq) technology has enabled the direct inference of co-expressions in specific cell types, facilitating our understanding of cell-type-specific biological functions. For this task, the high sequencing depth variations and measurement errors in scRNA-seq data present two significant challenges, and they have not been adequately addressed by existing methods. We propose a statistical approach, CS-CORE, for estimating and testing cell-type-specific co-expressions, that explicitly models sequencing depth variations and measurement errors in scRNA-seq data. Systematic evaluations show that most existing methods suffered from inflated false positives as well as biased co-expression estimates and clustering analysis, whereas CS-CORE gave accurate estimates in these experiments. When applied to scRNA-seq data from postmortem brain samples from Alzheimer's disease patients/controls and blood samples from COVID-19 patients/controls, CS-CORE identified cell-type-specific co-expressions and differential co-expressions that were more reproducible and/or more enriched for relevant biological pathways than those inferred from existing methods.

The past two decades have seen great advances in gene co-expression studies using microarrays and RNA-sequencing technologies, leading to rich insights on biological processes and disease mechanisms[1–3]. To date, most co-expression analyses have been performed on bulk samples that are a mixture of different cell types. As a result, the inferred networks are confounded with varying cell-type compositions across samples and limited to an aggregated view of gene regulations that may differ considerably across cell types[4,5]. To infer cell-type-specific networks from bulk samples, cell sorting can be performed, but the techniques are tedious and subject to technical artifacts[6].

With scRNA-seq technology such as droplet-based methods, gene expressions can now be measured in individual cells with annotated cell types[7], offering a great opportunity to construct cell-type-specific co-expression networks. However, such an analytical task is challenged by the unique characteristics of scRNA-seq data such as their high sequencing depth variations across cells and measurement errors. For

scRNA-seq data, the expression level of a specific gene is measured through the observed UMI (unique molecular identifier) count for this gene, and the sequencing depth of a cell is the sum of UMI counts across all genes. For a typical single cell experiment, there is substantial variation of sequencing depths across cells (e.g., 400–20,000)[8,9]. As a result, gene co-expressions measured via correlations of UMI counts across cells can be seriously confounded by varying sequencing depths, resulting in inflated false positive findings in detecting co-expressed gene pairs. This confounding issue cannot be addressed using standard normalization strategies, as will be shown later. Besides varying sequencing depths, measurement errors in the UMI count data pose an additional challenge in inferring co-expression levels as the errors tend to attenuate correlation estimates with different degrees for genes with different expression levels.

Several methods have been recently developed to better capture co-expressions from scRNA-seq data than a simple

[1]Department of Biostatistics, Yale University, New Haven, CT, USA. [2]Information Systems and Operations Management, Emory University, Atlanta, GA, USA. [3]Present address: Department of Biostatistics and Bioinformatics, Emory University, Atlanta, GA, USA. [4]Present address: Department of Mathematical Sciences, University of Cincinnati, Cincinnati, OH, USA. [5]Present address: Department of Biostatistics, University of Washington, Seattle, WA, USA. ✉e-mail: hongyu.zhao@yale.edu; emma.jzhang@emory.edu

normalization-based approach, including baredSC[10], locCSN[11], Noise Regularization[12], Normalisr[13], propr[14], and SpQN[15]. These methods consider different association metrics or additional adjustments when inferring co-expressions from scRNA-seq data. However, the proposed procedures do not have rigorous justifications as they are not explicitly based on the underlying data generating mechanisms, rely on restrictive distributional assumptions and do not appropriately account for measurement errors and varying sequencing depths across cells. Besides the above co-expression estimation methods, recently proposed methods such as sctransform[8] and analytic Pearson residuals[16] estimate gene expression levels from scRNA-seq data by removing the effect of varying sequencing depths via Pearson residuals under a negative binomial model. Although these two methods were not developed for co-expression estimation, one sensible approach is to calculate correlations of expression levels that have been adjusted for sequencing depths by either sctransform or analytic Pearson residuals; we refer to these approaches as $\rho$-sctransform and $\rho$-analytic PR, respectively, in our following discussion. As will be demonstrated later, the sequencing depth normalization in sctransform and analytic Pearson residuals, designed to infer expression levels, are inadequate in removing biases from sequencing depth variations and measurement errors when inferring co-expressions. In our systematic evaluations of different methods based on simulated and permuted real scRNA-seq data, we found that all the existing methods, including $\rho$-sctransform and $\rho$-analytic PR, suffer from inflated type-I errors, varying degrees of estimation biases, reduced power in detecting co-expressions, and potentially misleading results in downstream co-expression analysis such as clustering and principal component analysis.

Here, we present our proposed statistical approach for estimating and testing co-expressions from scRNA-seq data, called CS-CORE (cell-type-specific co-expressions). Specifically, CS-CORE models the unobserved true gene expression levels as latent variables, linked to the observed UMI counts through a measurement model that accounts for both sequencing depth variations and measurement errors. Under this model, CS-CORE implements a fast and efficient iteratively re-weighted least squares approach for estimating the true correlations between underlying expression levels, together with a theoretically justified statistical test to assess whether two genes are independent. The proposed model in CS-CORE does not impose any distributional assumptions on the underlying expression levels and can flexibly accommodate single cell data generating mechanisms such as negative binomial distributed counts. Through systematic evaluations based on simulated and permuted real scRNA-seq data, we found that CS-CORE had proper type-I error control, unbiased co-expression estimates and increased statistical power compared with other methods. CS-CORE also had satisfactory performance in downstream co-expression analysis.

We evaluated the utility of CS-CORE by applying it to multiple scRNA-seq data sets including postmortem brain samples from Alzheimer's disease patients and controls[17] and peripheral blood mononuclear cells (PBMC) of COVID-19 patients and controls[18]. For both diseases, CS-CORE identified co-expressions that were more reproducible across independent data sets and more enriched with known transcription factor-target gene pairs than other methods. Clustering analysis using results from CS-CORE extracted co-expressed and differentially co-expressed gene modules that were more strongly enriched for relevant cell-type-specific biological functions than those inferred from other methods, highlighting the potential of CS-CORE in characterizing cell-type-specific biological functions and uncovering disease-related cell-type-specific pathways.

## Results
### Overview of CS-CORE
We have $n$ cells from the same cell type with the observation for cell $i$, $i = 1, ..., n$, denoted by a vector $(x_{i1}, ..., x_{ip})$ corresponding to the observed UMI counts for $p$ genes. We use $s_i = \sum_{j=1}^{p} x_{ij}$ to denote the sequencing depth of cell $i$, which is the sum of UMI counts across all genes in this cell. Let $(z_{i1}, ..., z_{ip})$ denote the underlying expression levels from $p$ genes in cell $i$, defined to be the number of molecules from each gene relative to the total number of molecules in a cell[9]. Assume that:

$$(z_{i1}, ..., z_{ip}) \sim F_p(\boldsymbol{\mu}, \boldsymbol{\Sigma}), \quad x_{ij}|z_{ij} \sim \text{Poisson}(s_i z_{ij}), \tag{1}$$

where $F_p(\boldsymbol{\mu}, \boldsymbol{\Sigma})$ is an unknown nonnegative $p$-variate distribution with mean vector $\boldsymbol{\mu} = (\mu_1, ..., \mu_p)$, $\sum_{j=1}^{p} \mu_j = 1$, and covariance matrix $\boldsymbol{\Sigma} = (\sigma_{jj'})_{p \times p}$. Here, $x_{ij}$ is the UMI count of gene $j$ in cell $i$, assumed to follow a Poisson measurement model[9] depending on the underlying expression level $z_{ij}$ and sequencing depth $s_i$. This Poisson measurement model explicitly accounts for the sequencing depths and measurement errors. While a marginal expression-measurement model has been considered for modeling expression levels in bulk RNA-seq[19,20] and scRNA-seq data[8,21,22], a joint expression-measurement model such as Eq. (1) is needed to infer co-expressions. Under Eq. (1), if $z_{ij}$ follows a Gamma distribution, then $x_{ij}$ follows a negative binomial distribution marginally.

We measure gene co-expressions by $\boldsymbol{\Sigma}_{p \times p}$, which quantifies the correlation strength between the underlying expression levels, and $\boldsymbol{\Sigma}_{p \times p}$ is cell-type-specific as cells from the same cell type are considered (see Supplementary Discussion). This definition of co-expression is precise and not biased by sequencing depth variations and measurement errors. Specifically, for any gene pair $(j, j')$, we measure co-expression via their correlation $\rho_{jj'} = \sigma_{jj'} / \sqrt{\sigma_{jj} \sigma_{j'j'}}$.

Given UMI counts $\{x_{i1}, ..., x_{ip}\}_{i=1}^{n}$ and sequencing depths $\{s_i\}_{i=1}^{n}$, estimating the covariance matrix $\boldsymbol{\Sigma}_{p \times p}$ is challenging. Without placing distributional assumptions on $F_p$, we propose a moment-based iteratively reweighted least squares (IRLS) estimation procedure that is fast to implement and statistically efficient. For each gene pair $(j, j')$, we also develop a theoretically justified hypothesis testing procedure that evaluates the independence between their expression levels $z_{ij}$ and $z_{ij'}$. The test statistic can be easily computed using IRLS estimates, does not require distributional assumptions on $F_p$, and follows a standard normal distribution under the null. For all statistical tests performed in real data analyses, we applied a Benjamini-Hochberg (BH) procedure to control for the false discovery rate.

Details of the above estimation and testing procedures are given in Methods. In summary, CS-CORE takes UMI counts and sequencing depths across cells as input and estimates correlations of the underlying expression levels as well as $p$ values for testing independence between gene pairs, without needing parameter tuning. The procedure removes the confounding effects of varying sequencing depths and the bias from measurement errors when inferring co-expressions, is theoretically justified and fast to implement.

### CS-CORE has better control of false positive rates
To evaluate the performance of CS-CORE and illustrate the confounding effects from sequencing depth variations on other methods for independent gene pairs, we generated null data sets, where genes are not co-expressed, by permuting the single nucleus RNA-seq (snRNA-seq) data from ref. 17, while making the sequencing depths across cells either constant or varying. Specifically, we normalized gene expressions (UMI counts) within each cell by its sequencing depth and, for each gene, we randomly permuted its normalized expression levels across cells. Then, we obtained UMI counts for each gene based on pre-specified sequencing depth of each cell (Methods). To examine effects of sequencing depth variations, we considered two settings with one set to observed sequencing depths in real data, which are highly variable, and one set to be constant across cells. This permutation procedure de-correlated gene expressions such that the average co-expression for each gene in the permuted data, calculated

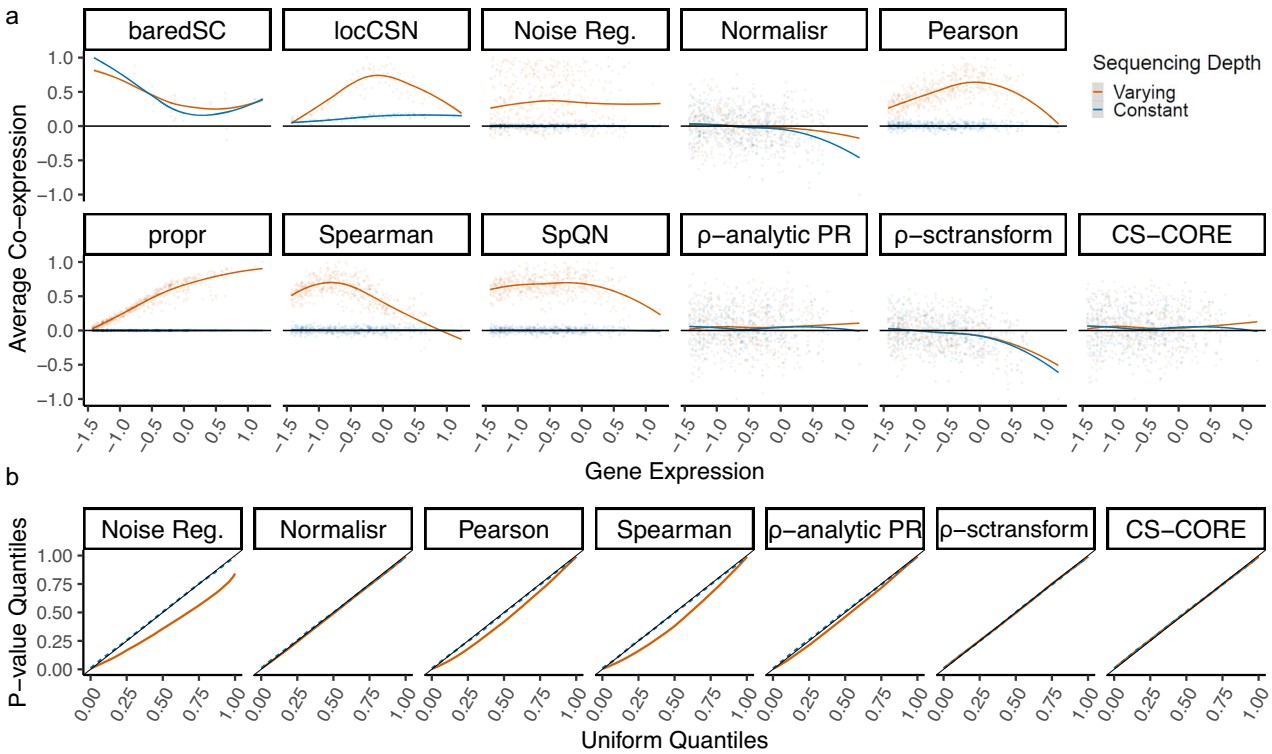

**Fig. 1 | Validation of CS-CORE using permuted snRNA-seq data from ref. 17.**
Results from permuted data with varying and constant sequencing depths are colored with light red and blue, respectively. **a** Scatter plots with fitted curves showing mean expression levels (*x*-axis) and average co-expression (*y*-axis) of each gene with co-expression estimated using baredSC, locCSN, Noise Regularization, Normalisr, Pearson correlation of log normalized data (Pearson), propr, Spearman correlation of log normalized data (Spearman), SpQN, $\rho$-analytic PR, $\rho$-sctransform and CS-CORE. Average co-expressions are re-scaled by the maximum value to aid comparison. The mean expression levels are plotted at the scale of $\log_{10}\mu_j + 3$ for $\mu_j$ defined in Eq. (1). **b** Q-Q plots comparing *p* values for testing co-expressions of gene pairs against Uniform(0,1) using seven methods with statistical tests, including Noise Regularization, Normalisr, Pearson correlation of log normalized data (Pearson), Spearman correlation of log normalized data (Spearman), $\rho$-analytic PR, $\rho$-sctransform and CS-CORE. Source data are provided as a Source Data file.

by averaging its co-expressions with all other genes, is expected to center around zero, regardless of sequencing depth variations.

We compared CS-CORE to other approaches, including baredSC[10], locCSN[11], Noise Regularization[12], Normalisr[13], Pearson correlation of log normalized data, propr[14], Spearman correlation of log normalized data, SpQN[15], $\rho$-analytic PR[16] and $\rho$-sctransform[8] (Methods). Among these approaches, statistical tests for co-expression are possible for Noise Regularization, Normalisr, Pearson correlation of log normalized data, Spearman correlation of log normalized data, $\rho$-analytic PR and $\rho$-sctransform.

For null data with high variations in sequencing depths, we found that co-expression estimates from most methods were biased with estimated average gene co-expressions different from zero (Fig. 1a). The amount of bias varied with the expression level with distinct patterns for different methods. Meanwhile, in null data with no sequencing depth variations, there were minimal biases for most methods (Fig. 1a), demonstrating that co-expression estimates can be biased by sequencing depth variations. By contrast, average co-expressions estimated by CS-CORE were unbiased and centered around zero, regardless of sequencing depth variations (Fig. 1a). We observed the same qualitative patterns in our experiments with simulated data (Supplementary Fig. 1). One main cause of bias from other methods is no or inadequate adjustments of sequencing depth variations when quantifying co-expressions, including the standard log transformations considered in locCSN, Pearson correlation of log normalized data, propr and Spearman correlation of log normalized data, as illustrated in Fig. 2, and post hoc adjustments considered in Noise regularization and SpQN. For baredSC, the bias may be due to reasons other than sequencing depth variations, such as violations of the

Gaussian mixture assumption on the underlying expression levels (Supplementary Notes). While Normalisr, $\rho$-analytic PR and $\rho$-sctransform are less confounded by sequencing depth variations, as they applied marginal regressions to explicitly adjust for sequencing depth variations, they had biases in estimating co-expressions (Fig. 3a). We also found that these three methods had reduced power in detecting co-expressions when compared to CS-CORE (Supplementary Fig. 2).

We also considered statistical tests for co-expressions in the permuted data. As the null hypothesis of no co-expression is expected to hold after permutation, *p* values for testing independence of gene pairs should follow the Uniform[0,1] distribution. In null data with no variations in sequencing depths, most methods had well-controlled type-I errors as the Q-Q plots showed matching quantiles between empirical distributions of *p* values and Uniform[0,1] (Fig. 1b). In null data with high variations in sequencing depths, Noise Regularization, Pearson, Spearman and $\rho$-analytic PR had inflated type I errors (Fig. 1b).

Next, we further explain why standard normalization procedures, such as scaling or log normalization, cannot remove the confounding when inferring co-expressions and illustrate via a simple experiment. We simulated UMI counts from an independent gene pair, each with a high mean expression level and follows a negative binomial distribution. We computed the scaled data as $10^4 \times x_i/s_i$ and the log normalized data as $\log(10^4 \times x_i/s_i + 1)$, and plotted the expression levels from gene 1 vs. gene 2 based on original, scaled and log normalized counts in Fig. 2. It is seen that spurious co-expression patterns appeared both for scaled and log normalized counts. The reason is as follows. Given two integers *a* and *b*, all cells with UMI counts *a*, *b* for these two genes, respectively, are plotted to the point (*a*, *b*) in the original UMI counts scatter plot. Interestingly, cells at this point, turning into

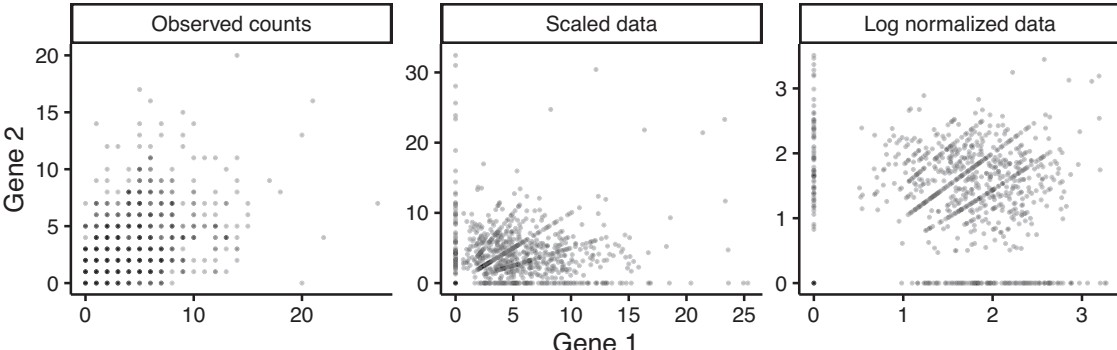

**Fig. 2 | Expressions of a simulated independent gene pair in original UMI counts, scaled and log normalized counts.** Gene 1 and gene 2 were simulated to have mean expression levels ranked 269 and 351 among 28,412 genes in excitatory neurons from ref. 17. We computed the scaled data as $10^4 \times x_i / s_i$ and the log normalized data as $\log(10^4 \times x_i / s_i + 1)$. Source data are provided as a Source Data file.

$(10^4 \times a/s_i, 10^4 \times b/s_i)$ after scaling, will be stretched out to form a line with a slope $b/a$ and an intercept 0 in the scaled data, and turning into approximately $(\log(a) - \log(s_i/10^4), \log(b) - \log(s_i/10^4))$ after log normalization, forming a line with a slope 1 and an intercept $\log(b) - \log(a)$ in the log normalized data (Fig. 2). These lines are artifacts of the normalization and can seriously inflate false positives when inferring co-expressions.

Using the UMI count data from this simulated independent gene pair in Fig. 2, we got the following co-expression estimates along with $p$ values (calculated for methods that offer tests) for existing methods that use log normalized data: locCSN = 0.61, Pearson correlation of log normalized data = 0.14 ($p$ value < 0.01), propr = 0.50, Spearman correlation of log normalized data = 0.07 ($p$ value = 0.02) and SpQN = 0.14.

## CS-CORE has better co-expression estimation accuracy and detection power

We evaluated the accuracy of CS-CORE in estimating and detecting co-expressions and illustrated another issue often referred to as the mean-correlation bias[15,23] in co-expression estimation. The mean-correlation bias is a separate issue from the confounding effect of varying sequencing depths. It arises, as measuring associations of the observed UMI counts, which profile the underlying expressions with measurement errors, tend to yield attenuated estimates due to the added errors. The amount of attenuation bias tends to decrease as the expression level increases (see Methods) and correlations tend to be more accurately estimated for highly expressed genes. As a result, highly expressed genes can appear more correlated as an artifact. This attenuation bias has also been noted in analyzing bulk RNA-seq data[15,24], but it can be exacerbated by the shallow sequencing depths frequently seen in scRNA-seq data.

To demonstrate this, we simulated expression data for gene pairs with varying expression levels and a correlation of $\rho = 0.5$ following marginal negative binomial distributions (see Methods). For co-expressed gene pairs with a true correlation of 0.5, we found that correlation estimates from all other methods were inaccurate (Fig. 3a) with most methods severely underestimating co-expressions for genes with low or moderate expression levels. The correlation estimates also spuriously increased with expression levels for most methods. By contrast, CS-CORE could accurately estimate co-expressions (Fig. 3a) and was not subject to mean-correlation bias. This is because CS-CORE is based on an expression-measurement model and explicitly measures co-expressions using correlations of the underlying expression levels, free of measurement errors. The mean-correlation bias remained on data simulated with no variations in sequencing depths (Supplementary Fig. 3), suggesting that the mean-correlation bias is a separate source of bias from varying sequencing depths. We also

evaluated the variances of different estimators and found that CS-CORE also achieved the smallest mean squared errors where both bias and variance were considered (Supplementary Fig. 4). We further evaluated the co-expression detection accuracy in simulations with $p = 500$ where co-expressed pairs were set to those inferred from real data (see Methods). The precision-recall curves in Fig. 3b show that CS-CORE achieves the highest area-under-the-curve value.

Finally, we compared the computing time of different methods (Fig. 3c) under the simulation setting considered in Fig. 3b. It is seen that CS-CORE is highly computationally efficient as it uses a least squares estimation procedure. Specifically, CS-CORE was faster to implement than the state-of-the-art method, locCSN, which is based a local nonparametric test and $\rho$-sctransform, which requires fitting marginal negative binomial regressions using likelihood-based approaches. The computing time of CS-CORE is comparable to simple procedures such as Pearson, Spearman and $\rho$-analytic PR, as these simple procedures all include a normalization step (see Methods).

## Other methods give biased results in downstream co-expression analysis

Bias in estimating co-expressions can negatively impact important downstream co-expression analyses such as clustering and principal component analysis (PCA). To evaluate the performance of CS-CORE and other methods on such downstream analytical tasks, we simulated $n = 2000$ cells for $p = 100$ genes with varying expression levels and a co-expression matrix with four co-expressed gene clusters (see Methods and Supplementary Methods). We estimated co-expression networks using CS-CORE and other methods, and compared them to the true co-expression network (Fig. 4a). In particular, when plotting the results from each method, we ordered the genes by applying hierarchical clustering to the estimated co-expression network. Estimated co-expression networks with the same gene ordering are shown in Supplementary Fig. 5. We found that CS-CORE was the only method that could accurately estimate co-expressions and be used to recover truly co-expressed gene clusters. The estimated co-expression networks and inferred cluster labels from other methods were strikingly inaccurate. These findings were further supported by evaluating the clustering accuracy (Fig. 4b), measured using adjusted Rand index, and the accuracy in estimating the top principal components (Fig. 4c), measured using subspace distance[25]. As a comparison, we simulated data with extremely high expression levels, so that measurement errors are much reduced, and with no sequencing depth variations and found that the clustering accuracy of other methods notably improved (Supplementary Fig. 6).

To highlight the mean-correlation bias, we computed the correlation between gene expression levels and estimated co-expression

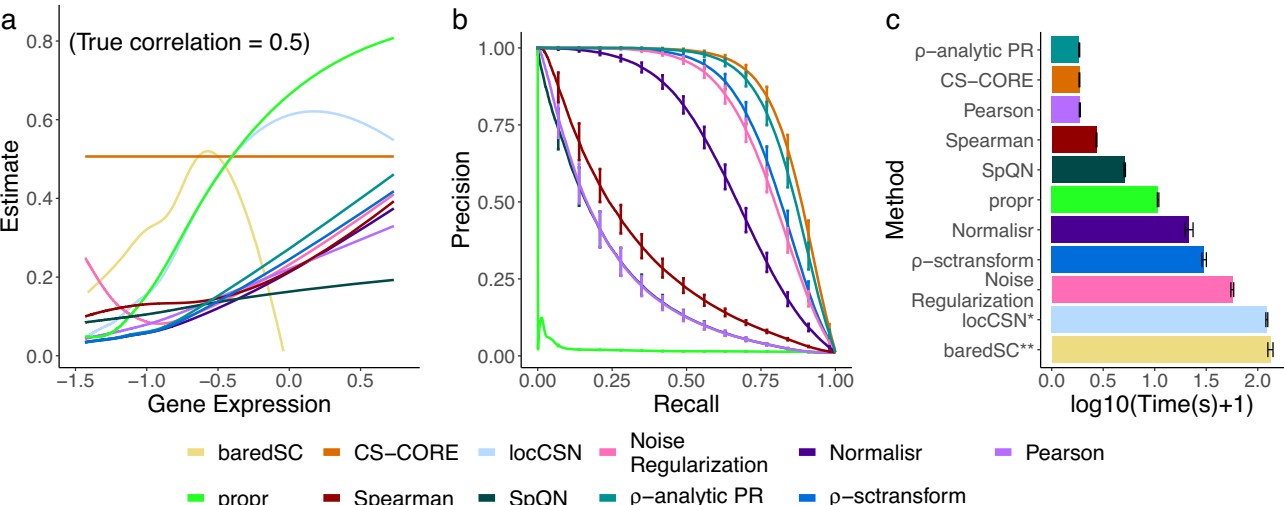

**Fig. 3 | Validation of CS-CORE using simulated data, compared to baredSC, locSCN, Noise Regularization, Normalisr, Pearson correlation of log normalized data (Pearson), propr, Spearman correlation of log normalized data (Spearman), SpQN, $\rho$-analytic PR and $\rho$-sctransform. a** Curve-fitted co-expression estimates against geometric mean expression levels on gene pairs simulated with a true correlation of 0.5 (5000 genes and 1000 cells). Mean expression levels were plotted at the same scale as in Fig. 1. **b** Precision-recall curves evaluated using 500 genes, 5000 cells and a sparse co-expression matrix estimated from real data. Cut-off values are based on $p$ values for CS-CORE, Noise Regularization, Normalisr, Pearson correlation of log normalized data, Spearman correlation of log normalized data, $\rho$-analytic PR and $\rho$-sctransform and absolute values of co-expression estimates for propr and SpQN, as they are not equipped with statistical tests; baredSC and locCSN were excluded due to their extreme demand for computing time. **c** Running times evaluated under the same setting as in (**b**). *locCSN is evaluated for all gene pairs using 0.2% of the cells to reduce computing time. **baredSC is evaluated using 0.2% of the cells and only for one gene pair. The simulations were run on an Intel Xeon Gold 6240 @ 2.60GHz with one node and 50 GB memory. The bars denote the average running times and the error bars denote one standard deviation across 100 replications. Source data are provided as a Source Data file.

levels. As expression levels were randomly assigned independent of correlation strengths, the true correlation between gene expression and co-expression levels should be close to zero, as marked in Fig. 4d. However, we found that the co-expression levels estimated from locCSN, Normalisr, Pearson correlation of log normalized data, propr, Spearman correlation of log normalized data, $\rho$-analytic PR and $\rho$-sctransform were spuriously correlated with the mean expression level. One implication of this mean-correlation bias is that, as highly expressed genes often appear highly co-expressed with other genes as an artifact, clustering methods tend to incorrectly cluster genes with similar expression levels in a co-expression cluster and expression levels become falsely predictive of the network modules (Fig. 4e). In another data example, we demonstrated that this mean-correlation bias could also lead to spurious clustering structures on null data where genes are not co-expressed (Supplementary Fig. 7).

## CS-CORE identified more biologically relevant co-expressions from AD samples

We applied CS-CORE to a snRNA-seq data set collected from the prefrontal cortical regions of 12 Alzheimer's disease (AD) patients and nine controls in ref. 17. We focused our comparison with $\rho$-sctransform, $\rho$-analytic PR and SpQN, as they give better overall performance in Figs. 1–4. First, using samples from controls, we estimated the co-expression network among top 5000 highly expressed genes in five major brain cell types including astrocyte (Ast), excitatory neuron (Ex), inhibitory neuron (In), oligodendrocyte (Oli) and microglia (Mic), and evaluated the reproducibility of identified co-expressions using two independent snRNA-seq data sets on prefrontal cortex from refs. 26,27 (Supplementary Methods). Figure 5a shows that the co-expressed gene pairs inferred by CS-CORE were more reproducible in ref. 26 than those inferred by $\rho$-sctransform across different $p$ value cutoffs and cell types, suggesting CS-CORE has greater statistical power to detect true co-expression signals. We had similar observations for data from ref. 27 (Supplementary Fig. 8) and for comparison with $\rho$-analytic PR (Supplementary Fig. 9A, B).

Next, by evaluating the overlap of co-expressed pairs with a database on known Transcription Factor(TF)-target gene pairs[28], we found CS-CORE recovered more known TF-target pairs than $\rho$-sctransform and $\rho$-analytic PR from the inferred networks (Fig. 5b and Supplementary Fig. 9C). Additionally, we extracted co-expressed gene modules by applying WGCNA[29] on significantly co-expressed gene pairs, which were then evaluated using Gene Ontology (GO) enrichment analysis[30] (see Supplementary Methods). Our enrichment analysis used the 5000 highly expressed genes as the background gene set, such that enrichment of any module is not attributed to its high expression levels. For microglia, the innate immune brain cells with a central role in the AD neuroinflammation mechanism[31], clustering based on CS-CORE identified four modules strongly enriched for GO terms related to microglia's functions, including defense response, chemical synaptic transmission, cytoplasmic translation and protein folding, respectively, while only two of these four functions were found enriched for modules inferred based on $\rho$-sctransform, and only one was found enriched based on SpQN and $\rho$-analytic PR, with less significant $p$ values and/or lower gene ratios (Supplementary Data 1). In particular, Fig. 5c shows the estimated co-expression networks, with genes ordered by hierarchical clustering, on a subset of genes from the four GO terms. It is seen that CS-CORE accurately grouped genes into respective biological functions, with genes in the same GO function densely connected. By contrast, $\rho$-sctransform only partially recovered some gene modules and the estimated co-expressions are generally much weaker, similarly for SpQN and $\rho$-analytic PR (Supplementary Fig. 10). Besides microglia, CS-CORE also identified gene modules that were enriched for cell-type-specific functions in astrocytes (synaptic signaling, protein folding, cellular response to hypoxia), inhibitory neurons (synaptic membrane) and oligodendrocytes (synaptic signaling, cholesterol metabolic process), while these functions were either not or much less enriched for modules inferred based on $\rho$-sctransform and $\rho$-analytic PR (Supplementary Data 1). When compared to SpQN, CS-CORE also identified more modules enriched for cell-type-specific functions across cell types, with the exception of a few

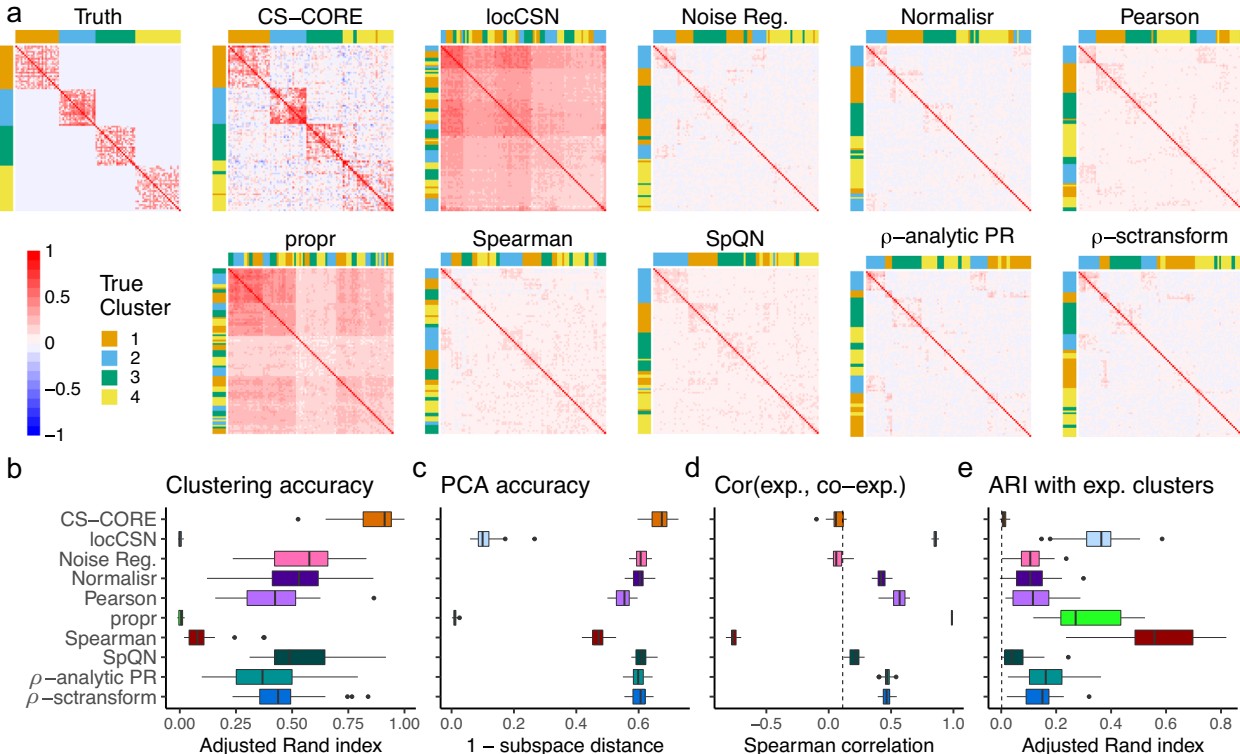

**Fig. 4 | Evaluation of CS-CORE in recovering co-expressed gene clusters and principal components of co-expression matrices using simulated data, compared to locCSN, Noise Regularization, Normalisr, Pearson correlation of log normalized data (Pearson), propr, Spearman correlation of log normalized data (Spearman), SpQN, $\rho$-analytic PR and $\rho$-sctransform. a** Heatmaps of true and estimated co-expression networks from simulations. When plotting results from each method, genes were ordered by applying hierarchical clustering to the estimated co-expression network and color coded by their true gene cluster labels. **b** Adjusted Rand index (ARI) between true co-expressed gene clusters and clusters extracted from co-expression networks estimated using different methods. The estimated clusters were obtained as described in (**a**) with the number of clusters set to 4. **c** Accuracy in recovering principal components, calculated using subspace

distance[25] between the top four singular vectors of the true co-expression matrix and those of the estimated co-expression matrix. **d** Spearman correlations between the expression levels and estimated average co-expression levels of genes, with ground truth calculated from simulation settings marked with a dashed line. **e** ARI between gene clusters extracted from estimated co-expression networks and gene clusters extracted from clustering gene expression levels, with the true ARI calculated from parameters used in simulation settings marked with a dashed line. **b**–**e** were evaluated with 25 replications. Data are presented as boxplots ($n = 25$ per group; center line, median; box limits, upper and lower quartiles; whiskers, up to 1.5 × interquartile range; points, outliers). Source data are provided as a Source Data file.

functions due to larger sizes of SpQN modules (Supplementary Notes). These results further highlight the potential of CS-CORE in uncovering cell-type-specific biological pathways.

Finally, we constructed the differential co-expression network in microglia between AD patients and controls from ref. 17 to investigate the biological pathways dysregulated in AD (see Methods). We applied clustering analysis to the differential network to extract gene modules that shared similar co-expression changes in AD and performed GO enrichment analysis. Clustering based on CS-CORE identified three differentially co-expressed gene modules enriched for cell-type-specific functional pathways that are implied in AD disease mechanisms, including protein folding[32], synapse signaling transduction[33], and protein kinase (toll-like receptors) signaling pathways[34] (Supplementary Data 2). In comparison, SpQN, $\rho$-analytic PR and $\rho$-sctransform did not identify any differentially co-expressed module enriched with cell-type-specific biological or disease-related functions (Supplementary Data 2).

### CS-CORE identified upregulated co-expressions from COVID-19 blood samples

We applied CS-CORE to a scRNA-seq data set from human peripheral blood mononuclear cells (PBMC) of seven hospitalized patients with SARS-CoV-2 and six controls[18] to identify biological pathways differentially regulated in COVID-19 patients.

Using samples from controls, we estimated cell-type-specific co-expressions among the top 5000 highly expressed genes in five major immune cell types, including B cells, CD4 positive T cells, CD8 positive T cells, monocytes and natural killer (NK) cells. Using an independent scRNA-seq data set on PBMC[35], we found that CS-CORE yielded a larger number of reproducible co-expressed gene pairs than $\rho$-sctransform and $\rho$-analytic PR across different $p$ value cutoffs and cell types (Supplementary Figs. 11A and 12A). CS-CORE also uncovered more gene pairs that overlapped with known TF-target gene pairs and more gene modules with stronger cell-type-specific functional enrichment than $\rho$-sctransform and $\rho$-analytic PR across cell types through GO enrichment analysis (Supplementary Figs. 11B and 12B and Supplementary Data 3). For example, CS-CORE identified three co-expression modules enriched for the biological functions of B cells, including antigen processing via MHC Class II, adaptive immune response and response to inteferon-alpha (Supplementary Data 3). In contrast, only one of these three functions was found enriched in a module inferred based on $\rho$-sctransform with a less significant $p$ value and a lower gene ratio, similarly for SpQN and $\rho$-analytic PR (Supplementary Data 3). Our results on PBMC again show that CS-CORE can recover biologically more meaningful co-expressions than other methods.

We next investigated cell-type-specific responses to SARS-CoV-2 viral infection in monocytes using a differential co-expression analysis similar to the one performed in the previous section between AD

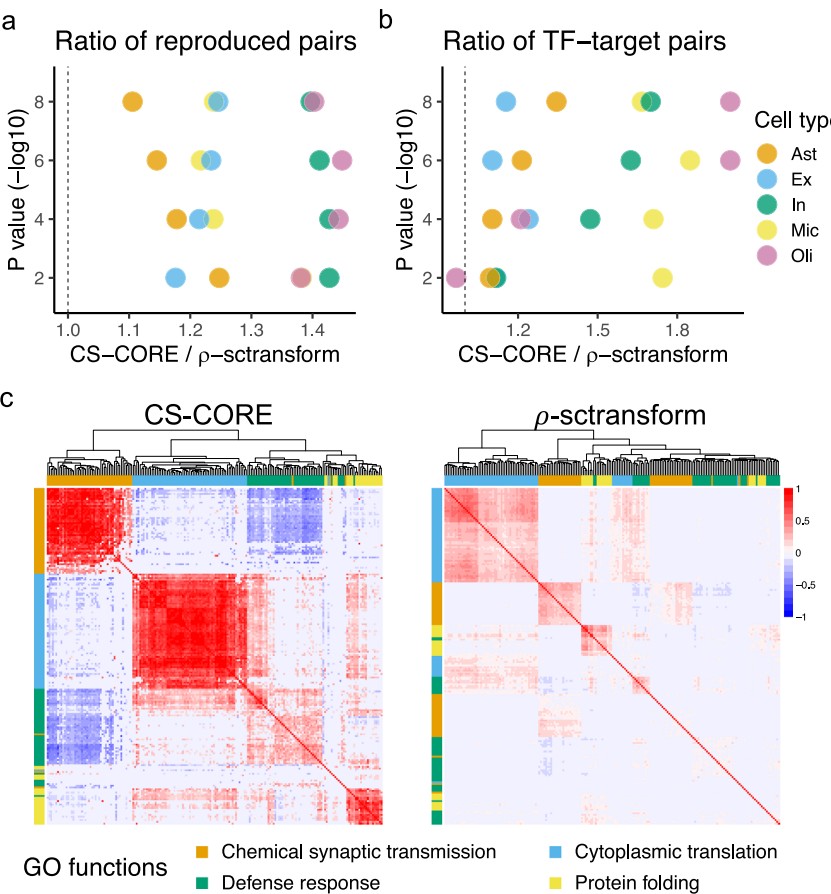

**Fig. 5 | Co-expression analysis using AD brain samples in ref. 17.** We used the cells in five major brain cell types from control subjects from ref. 17 to estimate cell-type-specific co-expression networks. **a** Ratio of the numbers of gene pairs that were identified as significant in both refs. 17,26 at specified *p* value cutoffs between CS-CORE and *ρ*-sctransform. **b** Ratio of the numbers of gene pairs that were identified as significant and overlapped with known TF-target gene pairs in the TRRUST database[28] between CS-CORE and *ρ*-sctransform. **c** Heatmaps of microglia-specific co-expression network estimates on genes from four GO terms on microglia's functions with genes ordered by hierarchical clustering. In (**a**, **b**), *p* values were evaluated based on two-sided tests described in Methods and nominal *p* values not adjusted for multiple testing were used to determine statistical significance. Source data are provided as a Source Data file.

patients and controls. Clustering analysis revealed gene modules that share similar co-expression changes in monocytes in response to SARS-CoV-2. In particular, four gene modules inferred using co-expression estimates from CS-CORE were significantly enriched for immune responses based on GO enrichment analysis, including virus defense response, antigen processing, leukocyte mediated immunity and cellular stress response (Supplementary Data 4). In contrast, *ρ*-sctransform modules missed the functions of antigen processing and leukocyte mediated immunity and SpQN and *ρ*-analytic PR did not identify any gene module associated with immune responses (Supplementary Data 4). In Fig. 6, we highlight a module identified by CS-CORE, which is enriched for the interferon signaling pathway (Supplementary Table 1), a key immune signature in COVID-19 patients that has been demonstrated in multiple studies[36–38]. While it is known that the expression levels of interferon-stimulated genes are upregulated in monocytes from COVID-19 patients, by comparing the CS-CORE estimates in monocytes between COVID-19 patients and controls, we identified upregulated co-expressions among interferon-stimulated genes, suggesting increased gene coordination in the interferon signaling pathway upon viral infection. We also found stronger co-expressions between genes in the interferon signaling and antigen presentation pathways among COVID-19 patients, suggesting stronger concerted immune responses between these two pathways. Finally, we note that this gene module also contains multiple known genes in the SARS-CoV-2 infection Reactome pathway, revealing cell-type-specific changes in co-expressions among known disease-related genes.

## Discussion

We developed a comprehensive statistical approach, CS-CORE, for estimating and testing cell-type-specific co-expressions based on scRNA-seq data. CS-CORE adopts a multivariate expression-measurement model for the observed UMI counts and a pair-wise IRLS method for estimation and testing. It does not place distributional assumptions on the underlying expression levels and can be implemented very efficiently to estimate and test co-expressions in a large network. We demonstrated the better performance of CS-CORE than other methods through both simulations and real data analyses.

Our work pointed to two potential sources of biases when inferring co-expressions from UMI counts. The first one is the varying sequencing depths across cells, which can lead to inflated false positive findings in detecting co-expressions, as a pair of independent genes may appear co-expressed as a result of the sequencing depth variations across cells. The second one is the error from the measurement process, causing the observed UMI counts to deviate from the underlying expression levels. Under the Poisson measurement model, this deviation is a function of both the expression level and the sequencing depth. When estimating the underlying co-expression level for a pair of genes, correlations between UMI counts tend to be biased toward zero as a result of the measurement errors. In our experiments, we observed such an attenuation bias in most methods we compared to, leading to inaccuracy and reduced power in estimating and detecting co-expressions. These two distinct sources of biases, when combined, cause serious issues in estimating and testing

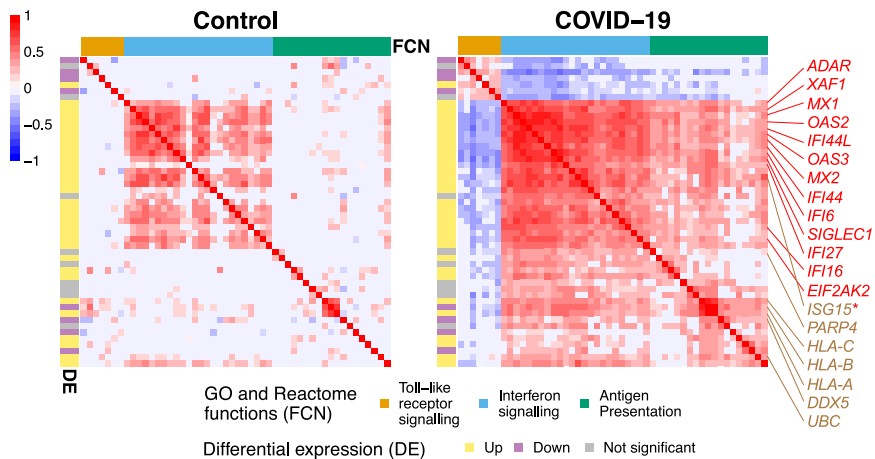

**Fig. 6 | CS-CORE estimates in monocytes from control subjects and COVID-19 patients.** Known interferon-stimulated genes are colored in red. Genes in the SARS-CoV-2 infection Reactome pathway are colored in brown. * is used to mark genes that belong to both gene sets. We performed a differential co-expression analysis on top 1000 highly expressed genes in monocytes and obtained modules of genes that shared similar changes in co-expressions between cells from COVID-19 patients and controls. For a differentially co-expressed gene module enriched for the interferon signaling pathway, we focused on genes that had strong differential signals (sum of absolute differential co-expressions greater than the median) and visualized the co-expression network estimates in control subjects and COVID-19 patients. Source data are provided as a Source Data file.

for co-expressions. As demonstrated in our analysis, no other methods can adequately address both. Our approach CS-CORE addresses them by explicitly modeling the measurement process, accounting for both varying sequencing depths and measurement errors, and estimates the first and second moments of the underlying multivariate expression model to produce estimates of co-expressions, without any specific distributional assumptions.

There has been recent work that makes cell-type-specific inferences from bulk samples leveraging cell-type deconvolution techniques[39,40]. These work often aims to estimate cell-type-specific expressions and compositions in bulk samples[41–44]. In particular, a recent method CSNet[5] focuses on estimating cell-type-specific co-expressions from bulk sample data. The rich bulk samples collected over past decades and the increasingly available scRNA-seq data together offer a great opportunity to integrate bulk samples and single cell data to draw cell-type-specific inferences of co-expressions. The proposed method CS-CORE provides a useful tool in developing methods for such integrative analyses.

In CS-CORE, we use Poisson distribution for the measurement model in CS-CORE, as it agrees with the existing literature[8,9,16] that a Poisson distribution is usually sufficient to characterize the variations introduced by the sequencing experiment. It can be useful to adapt CS-CORE to model $x_{ij}|z_{ij}$ using other distributions that model nonnegative integers such as the negative binomial distribution and we leave it to future work.

In CS-CORE, we have assumed that gene expressions from cells of the same cell type follow the same distribution. This assumption may not hold when the cells are collected from individuals with different genetic, demographic and clinical characteristics. For example, there is a growing interest in studying the genetic basis of cell-type-specific gene expression and co-expression differences across individuals using single cell data, and such population level single cell data are becoming increasingly available[45,46]. As an important next step, we plan to extend the CS-CORE framework to infer individualized cell-type-specific co-expression networks and to study the differences in gene co-expressions across genotypes and conditions, shedding light on individualized and context-specific biological functions and pathways.

In summary, the CS-CORE method introduced in this article is statistically sound and computationally efficient. Compared to the other methods, it generates more reproducible and biologically more

relevant cell-type-specific co-expression networks across multiple scRNA-seq data sets. With the rapid increase of scRNA-seq studies, we believe that CS-CORE offers a powerful and robust statistical tool to infer cell-type-specific co-expression networks to characterize biological pathways and molecular mechanisms at the cell type level.

## Methods
### CS-CORE method
Under the expression-measurement model defined in Eq. (1), it holds that $\mathbb{E}(x_{ij}) = s_i\mu_j$, $\mathrm{Var}(x_{ij}) = s_i\mu_j + s_i^2\sigma_{jj}$, and $\mathbb{E}[(x_{ij} - s_i\mu_j)(x_{ij'} - s_i\mu_{j'})] = s_i^2\sigma_{jj'}$. This motivates us to estimate $\mu_j$'s and $(\sigma_{jj'})_{p\times p}$ via the following set of regression equations:

$$
\begin{aligned}
x_{ij} &= s_i\mu_j + \epsilon_{ij}, \\
(x_{ij} - s_i\mu_j)^2 &= s_i\mu_j + s_i^2\sigma_{jj} + \eta_{ij}, \\
(x_{ij} - s_i\mu_j)(x_{ij'} - s_i\mu_{j'}) &= s_i^2\sigma_{jj'} + \xi_{ijj'},
\end{aligned}
\tag{2}
$$

where $\epsilon_{ij}$, $\eta_{ij}$, and $\xi_{ijj'}$ are independent and mean-zero error variables for all $i, j, j'$. Specifically, given UMI counts $x_{ij}$'s and sequencing depths $s_i$'s, the mean parameter $\mu_j$ is estimated via $\min_\mu \sum_{i=1}^n w_{ij}(x_{ij} - s_i\mu)^2$, where $w_{ij}$ is the weight for cell $i$ to be determined. Given the estimates $\hat\mu_j$'s, we estimate $\sigma_{jj}$ and $\sigma_{jj'}$ with $\min_\sigma \sum_{i=1}^n h_{ij}[(x_{ij} - s_i\hat\mu_j)^2 - s_i\hat\mu_j - s_i^2\sigma]^2$ and $\min_\sigma \sum_{i=1}^n g_{ijj'}[(x_{ij} - s_i\hat\mu_j)(x_{ij'} - s_i\hat\mu_{j'}) - s_i^2\sigma]^2$, respectively, where $h_{ij}$ and $g_{ijj'}$ are weights to be determined. These weighted least squares can be computed very efficiently.

In CS-CORE, we carefully select and update the weights via an IRLS procedure, such that the weighted least squares estimators are statistically efficient. The most ideal weights, in terms of minimizing the variance of the IRLS estimator, should be the reciprocal of the variances of the error variables in Eq. (2)[47]. Hence, we set $w_{ij} = 1/\mathrm{Var}(\epsilon_{ij}) = 1/(s_i\mu_j + s_i^2\sigma_{jj})$, which is updated in each step of the IRLS estimation. The analytical forms of $\mathrm{Var}(\eta_{ij})$ and $\mathrm{Var}(\xi_{ijj'})$ are difficult to derive as we do not place distributional assumptions on $z_{ij}$. Given weights $w_{ij}$'s for the mean parameter estimation, we set weights for variance and covariance estimation as $h_{ij} = w_{ij}^2$ and $g_{ijj'} = w_{ij}w_{ij'}$, respectively, which yield good performance in our experiments (Supplementary Notes) and the IRLS procedure typically converges within five iterations. In practice, we add a regularization step to the variance parameters $\sigma_{jj}$'s used in calculating the weights, as their

estimates can be variable, leading to highly variable weights. Specifically, we wrote $\sigma_{jj} = \mu_j^2 \times \theta_j$ and regularized the over-dispersion parameter $\theta_j$ across genes, inspired by a similar idea in DESeq2[20] and sctransform[8]. We found that such a simple regularization step leads to stable weight estimates and a reduced variance of the weighted least squares estimator. The detailed procedure for parameter estimation is presented in Algorithm 1, where IRLS formulas $f_{\mu_j}(\cdot), f_{\sigma_{jj}}(\cdot), f_{\sigma_{jj'}}(\cdot)$ for estimating $\mu_j$, $\sigma_{jj}$ and $\sigma_{jj'}$ can be found in Supplementary Methods.

Next, we develop a statistical test to assess whether a gene pair have independent expression levels. Under the model in Eq. (1) and when $z_{ij}$ and $z_{ij'}$ are independent, $\mathrm{Var}(\xi_{ijj'}) = (s_i\mu_j + s_i^2\sigma_{jj})(s_i\mu_{j'} + s_i^2\sigma_{j'j'}) = 1/g_{ijj'}$. Letting $\hat{\sigma}_{jj'}$ be estimated with true $\mu_j$'s, we define the test statistic $T_{jj'} = \hat{\sigma}_{jj'}/\sqrt{\mathrm{Var}(\hat{\sigma}_{jj'})}$, which can be calculated as:

$$T_{jj'} = \frac{\sum_i s_i^2(x_{ij} - s_i\mu_j)(x_{ij'} - s_i\mu_{j'})g_{ijj'}}{\sqrt{\sum_i s_i^4(s_i\mu_j + s_i^2\sigma_{jj})(s_i\mu_{j'} + s_i^2\sigma_{j'j'})g_{ijj'}^2}}.$$

It then follows that $T_{jj'} \sim N(0,1)$ under the null hypothesis that $z_{ij}$ and $z_{ij'}$ are independent. This result allows us to directly compute $p$ values by plugging in IRLS estimated $\mu_j$'s and $\sigma_{jj}$'s, all of which are consistent weighted least squares estimators.

**Algorithm 1**. CS-CORE estimation

1: **Input:** UMI counts $X = (x_{ij})_{n \times p}$ with $n$ cells and $p$ genes, sequencing depths $\{s_i\}_{i=1}^n$
2: Set $\Delta^{(0)} = 1$. Set $t = 0$.
3: // Estimate $\mu_j$'s and $\sigma_{jj}$'s
4: // Initialize with ordinary least sqaures
5: **for** $j = 1, \ldots, p$ **do**
6: $\hat{\mu}_j^{(t)} \leftarrow f_{\mu_j}(1)$, $(\hat{\sigma}_{jj})^{(t)} \leftarrow f_{\sigma_{jj}}(\hat{\mu}_j^{(t)}, 1)$.
7: **end for**
8: // Iteratively reweighted least squares
9: **while** $\Delta^{(t)} \geq 0.05$ **do**
10: $t \leftarrow t + 1$
11: // Regularize $\theta_j$ estimates for weighting
12: $\hat{\theta}^{(t)} \leftarrow \mathrm{median}_j\{(\hat{\sigma}_{jj})^{(t-1)}/(\hat{\mu}_j^{(t-1)})^2\}$
13: // Update $\mu_j$, $\sigma_{jj}$ estimates
14: **for** $j = 1, \ldots, p$ **do**
15: $w_{ij}^{(t)} \leftarrow 1/[s_i\hat{\mu}_j^{(t-1)} + s_i^2(\hat{\mu}_j^{(t-1)})^2 \times \hat{\theta}^{(t)}]$
16: $\hat{\mu}_j^{(t)} \leftarrow f_{\mu_j}(w_{ij}^{(t)})$
17: $h_{ij}^{(t)} \leftarrow 1/[s_i\hat{\mu}_j^{(t)} + s_i^2(\hat{\mu}_j^{(t)})^2 \times \hat{\theta}^{(t)}]^2$
18: $(\hat{\sigma}_{jj})^{(t)} \leftarrow f_{\sigma_{jj}}(\hat{\mu}_j^{(t)}, h_{ij}^{(t)})$
19: **end for**
20: // Assess convergence
21: $\Delta^{(t)} \leftarrow \max_j |\log(\hat{\sigma}_{jj})^{(t)} - \log(\hat{\sigma}_{jj})^{(t-1)}|$
22: **end while**
23: // Estimate $\sigma_{jj'}$'s
24: Let $\hat{\theta} = \hat{\theta}^{(t)}$, $\hat{\mu}_j = \hat{\mu}_j^{(t)}$, $\hat{\sigma}_{jj} = (\hat{\sigma}_{jj})^{(t)}$ for $j, j' = 1, \ldots, p$.
25: **for** $j, j' = 1, \ldots, p, j \neq j'$ **do**
26: $g_{ijj'} = 1/\{[s_i\hat{\mu}_j + s_i^2(\hat{\mu}_j)^2 \times \hat{\theta}][s_i\hat{\mu}_{j'} + s_i^2(\hat{\mu}_{j'})^2 \times \hat{\theta}]\}$
27: $\hat{\sigma}_{jj'} = f_{\sigma_{jj'}}(\hat{\mu}_j, \hat{\mu}_{j'}, g_{ijj'})$
28: $\hat{\rho}_{jj'} = \hat{\sigma}_{jj'}/\sqrt{\hat{\sigma}_{jj}\hat{\sigma}_{j'j'}}$
29: **end for**
30: **Output:** $\hat{\mu}_j, \hat{\sigma}_{jj}, \hat{\sigma}_{jj'}$ for $j, j' = 1, \ldots, p$

## Other co-expression estimation and testing methods using scRNA-seq data

We compared CS-CORE with ten other methods for inferring gene co-expression from single cell data, including baredSC[10], locCSN[11], Noise Regularization[12], Normalisr[13], Pearson correlation of log normalized data, propr[14], Spearman correlation of log normalized data, SpQN[15], $\rho$-analytic PR[16] and $\rho$-sctransform[8]. The method baredSC was evaluated with the implementation provided at https://baredsc.readthedocs.io/en/latest/ with default parameters. The method locCSN was applied on log normalized data $\log(10^4 \times x_{ij}/s_i + 1)$ and computed with the Python implementation provided at https://github.com/xuranw/locCSN. While locCSN estimates one network per cell, we followed the authors' instructions to aggregate cell-specific co-expressions into cell-type-specific co-expressions, as stated in Wang et al.[11] that averaging provides stable estimates of the network structure. The method propr refers to $\rho_p$ in ref. 14 and was calculated with the R package "propR" (v.4.2.6). For $\rho$-analytic PR, we computed the analytic Pearson residuals as described in ref. 16 and evaluated Pearson correlations between the residuals. For $\rho$-sctransform, we computed the residuals of sctransform using R package Seurat (v.4.0.3) and evaluated Pearson correlations between the residuals. The Spearman (Pearson) correlation was calculated on log normalized expression data $\log(10^4 \times x_{ij}/s_i + 1)$ using the R package "stats" (v.4.1.3). Noise Regularization[12] was implemented from https://github.com/RuoyuZhang/NoiseRegularization, Normalisr[13] was computed with the Python implementation from https://github.com/lingfeiwang/normalisr (v.1.0.0) and SpQN[15] was computed with R package "SpQN" (v.1.6.0).

Among the above ten methods, statistical tests for co-expressions are available for Noise Regularization, Normalisr, Pearson correlation of log normalized data, Spearman correlation of log normalized data, $\rho$-analytic PR and $\rho$-sctransform. Specifically, the $p$ values for Normalisr were computed using the online code provided for its implementation. Test statistics for all other methods with statistical tests were calculated as $t = r\sqrt{(n-2)/(1-r^2)}$ given the correlation estimate $r$, and two-sided $p$ values were evaluated under the standard normal distribution. For methods that do not offer statistical tests or suffer from inflated type-I errors, we evaluated empirical $p$ values using a simulation-based approach (Supplementary Methods).

## Experiments with permuted scRNA-seq data

To generate null data sets from a given scRNA-seq data set with co-expression levels at or close to zero among all gene pairs while preserving gene expression levels, we adopt the following approach that combines permutation with Poisson sampling. First, we calculated normalized expression level for each gene $j$ in cell $i$, written as $y_{ij} = x_{ij}/s_i$. Then, for each gene $j$, we randomly permuted the normalized expressions $(y_{ij})_{i=1,\ldots,n}$ across $n$ cells. After permutation, gene expressions were decorrelated and no gene pairs were expected to co-express. Finally, the UMI count of gene $j$ from cell $i$ in the permuted data was calculated by sampling from $\mathrm{Poisson}(t_i y_{ij}^p)$, where $y_{ij}^p$ is the normalized expression level after permutation and $t_i$ is the desired sequencing depth in cell $i$. For the varying and constant sequencing depth settings in Fig. 1, we set $t_i$ to the observed sequencing depth $s_i$ and median$(s_1, \ldots, s_n)$, respectively.

For numerical results in Fig. 1 and Supplementary Fig. 1, we used the snRNA-seq data from ref. 17 and selected excitatory neurons from control subjects. The distribution of sequencing depths is long-tailed with a median of 5833 (Supplementary Fig. 13). We randomly sampled 1000 cells and their corresponding sequencing depths. For all methods except for baredSC and locCSN, we focused on 500 genes randomly sampled from the top 5000 highly expressed genes with probabilities proportional to the inverse density of expression levels. This ensures that the sampled genes could cover the range of expression levels. For baredSC and locCSN, we focused on 20 and 100 genes randomly sampled from the top 5000 highly expressed genes in a similar way respectively due to their extreme computational costs (Fig. 3c). For numerical results in Fig. 1b, we further repeated permutations for 100 times, randomly selected 100 gene pairs and used their $p$ values from all replicates to make Q-Q plots.

## A simple illustration of the expression-level-dependent attenuation bias

To illustrate how errors from the Poisson measurement model in Eq. (1) can bias co-expression estimates, we conduct a short analysis under a much simplified case that directly calculates Pearson correlations of UMI counts. The analysis is similar to that in ref. 15, though $s_i$ was not considered there. From Eq. (1) and for genes $j, j'$, we have:

$$\frac{E([(x_{ij}-E(x_{ij})][x_{ij'}-E(x_{ij'})])}{\sqrt{\text{Var}(x_{ij})\text{Var}(x_{ij'})}} = \rho_{jj'} \times a_{ij}a_{ij'},$$

$$a_{ij} = \sqrt{\frac{s_i^2\text{Var}(z_{ij})}{\text{Var}(x_{ij})}} = \sqrt{\frac{s_i\text{CV}_j^2}{1/\mu_j + s_i\text{CV}_j^2}}, \quad (3)$$

where $\text{CV}_j$ is the coefficient of variation of gene $j$ defined as $\sqrt{\sigma_{jj}}/\mu_j$. To measure the true correlation $\rho_{jj'}$, the correlation based on UMI counts $x_{ij}$ and $x_{ij'}$ is always biased toward zero, as $a_{ij}a_{ij'}<1$ when $\mu_j, \mu_j'>0$. We refer to $a_{ij}$, derived under the Poisson measurement model in Eq. (1), as the attenuation factor in this analysis.

When $\text{CV}_j$'s are fixed, the attenuation factor $a_{ij}$ is closer to 1 for highly expressed genes with a larger $\mu_j$. Correspondingly, correlations are more accurately estimated for highly expressed genes and more attenuated for lowly expressed genes, assuming $s_i$'s do not vary across cells. Based on a real snRNA-seq data set from ref. 17, we indeed observed that the estimated $a_{ij}$ approached 1 as the gene expression level increased (Supplementary Fig. 14). With $s_i$'s varying across cells, the UMI counts for a pair of genes across cells are not identically distributed. In this case, it is difficult to analytically demonstrate the combined effect of the attenuation bias and the varying sequencing depths on co-expression estimation.

## Simulating from the multivariate expression-measurement model

To simulate gene expression data from the model in (1), we combine a marginal negative binomial model and a copula-based approach that can simulate multivariate count data following a pre-specified co-expression matrix. All simulation experiments were designed to simulate cells from the same cell type.

We specified the distribution of true expression level $z_{ij}$ to be Gamma$(\alpha_j, \beta_j)$ where $\mu_j = \alpha_j\beta_j$ and $\sigma_{jj} = \alpha_j\beta_j^2$ correspond to the marginal mean and variance in Eq. (1). Conditional on $z_{ij}$, we simulated counts $x_{ij}$ from Poisson$(s_i z_{ij})$ independently for cell $i$ and gene $j$. Marginally, this Poisson-Gamma mixture is equivalent to a negative binomial model on $x_{ij}$, which is commonly used to model droplet-based single cell data[8,48–50]. In our simulations, $\mu_j$, $\sigma_{jj}$ and $s_i$ are estimated or sampled from real data (see Supplementary Methods). Next, given a $p \times p$ correlation matrix $R$, we adopted a Gaussian copula to simulate correlated Gamma random variables[51,52]. In particular, we first simulated samples $(v_{i1}, ..., v_{ip})$ from a multivariate normal distribution with mean 0 and correlation $R$ and then computed $z_{ij} = F_j^{-1}(\Phi(v_{ij}))$, where $\Phi(\cdot)$ is the cumulative distribution function (CDF) of a standard normal distribution and $F_j(\cdot)$ is the CDF of Gamma$(\alpha_j, \beta_j)$. In Fig. 3b, the matrix $R$ was estimated from ref. 17 and in Fig. 4a, the modular matrix $R$ was generated from a network model. These details can be found in Supplementary Methods.

## Differential co-expression analysis

For differential co-expression analysis, we first estimated co-expression networks from the disease and control groups separately. For the group with more cells, we randomly sampled a subset of cells such that the two groups had the same number of cells when estimating co-expressions. For each gene pair, we calculated the difference between co-expression estimates and assessed the statistical significance using a permutation test, where we randomly permuted the group labels 100 times and built a null distribution of differences in co-expressions. We then applied WGCNA[29] to the significantly differentially co-expressed pairs (BH-adjusted $p$ values <

### Table 1 | Summary of single cell (nucleus) RNA-seq data used in analyses

| Data sets | 17 | 26 | 27 | 18 | 35 |
|---|---|---|---|---|---|
| Tissue | Brain | Brain | Brain | PBMC | PBMC |
| Data | snRNA-seq | snRNA-seq | snRNA-seq | scRNA-seq | scRNA-seq |
| #cells/ nuclei | 169,500 | 70,634 | 61,472 | 44,721 | 153,554 |
| #cell types | 6 | 8 | 7 | 13 | 29 |
| Median seq depth | 2600 | 1474 | 6382 | 1946 | 3618 |
| #genes | 28,412 | 17,926 | 36,114 | 26,361 | 33,538 |
| #samples | 21 | 48 | 18 | 14 | 31 |
| Accession codes | GSE157827 | syn21261143 | syn22079621 | PBMCs from Blish lab | GSE155224 |

The following information is provided for each data set: reference, tissue, data type (scRNA-seq/snRNA-seq), number of cells/nuclei, number of cell types, median sequencing depths, number of genes, number of samples and accession codes.

0.05) with the soft-thresholding power set to 1 and extracted differentially co-expressed modules.

## Data summary and pre-processing

A summary of the data sets analyzed in our work is given in Table 1. For cell-type labels of the cells, we used the cell-type labels provided by authors of refs. 26,27,18,35. Cell-type labels were not provided for the data set from ref. 17 and we annotated the cell types following the procedure described in ref. 17.

To conduct the reproduciblity analysis in five major immune cell types between ref. 18 and ref. 35, we combined the Naive B cells and Memory B cells from ref. 35 to compare with B cells from ref. 18; we combined the CD14 Monocytes and CD16 Monocytes from ref. 18 to compare with the combination of classical monocytes and NC and IM monocytes from ref. 35; we combined the Naive CD4 T, Memory CD4 T and Memory CD4 and MAI T cells from ref. 35 to compare with CD4 T cells from ref. 18; we combined the Memory CD8 T, Naive CD8 T, Effector T and IFN-activated CD8 T cells from ref. 35 to compare with CD8 T cells from ref. 18; we combined the NK CD56dim and NK CD56bright cells from ref. 35 to compare with the NK cells from ref. 18.

## Statistics and reproducibility

This study used data from published studies and details on the study design can be found in the original publications listed in Table 1. The statistical analysis of the data was described in Methods. The results can be reproduced following Methods. No statistical method was used to predetermine sample size. No data were excluded from the analyses. The experiments were not randomized. The investigators were not blinded to allocation during experiments and outcome assessment.

## Reporting summary

Further information on research design is available in the Nature Portfolio Reporting Summary linked to this article.

## Data availability

All data used in this work are publicly available, including GSE157827, syn21261143, syn22079621, COVID-19 Peripheral Blood Mononuclear Cells (PBMCs) (https://www.covid19cellatlas.org/index.patient.html), and GSE155224. More details on these data are included in Table 1. For functional enrichment analysis, we used the Gene Ontology Database provided by R package clusterProfiler (v.4.2.2)[30] and the Reactome Pathway Database provided by R package ReactomePA (v.1.38.0)[53].

Additional raw data for producing figures have been deposited in Zenodo under accession code 7983559[54]. Source data are provided with this paper.

## Code availability

Codes that implement CS-CORE are covered by the MIT License and are available on GitHub (https://github.com/ChangSuBiostats/CS-CORE), (https://github.com/ChangSuBiostats/CS-CORE_python) and Zenodo (https://doi.org/10.5281/zenodo.7983426)[55]. Tutorials on usage are also provided (https://changsubiostats.github.io/CS-CORE/).

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

## Acknowledgements

J.Z.'s research is supported by NSF DMS-2015190 and DMS-2210469. B.C.'s research is supported in part by DOD W81XWH2110019. C.S., Z.X., X.S., and H.Z.'s research is supported in part by NIH R01 GM134005 and R56 AG074015.

## Author contributions

C.S., Z.X., H.Z. and J.Z. designed research; C.S., Z.X. and X.S. performed research and analyzed data; C.S., Z.X., X.S., B.C. and J.Z. contributed analytic tools; C.S., Z.X., X.S., H.Z. and J.Z. wrote the paper; H.Z. and J.Z. jointly supervised the work.

## Competing interests

The authors declare no competing interests.
