## [Peer Review File · Nature Communications]

Cell-type-specific co-expression inference from single cell RNA-sequencing dataREVIEWER COMMENTS

Reviewer #1 (Remarks to the Author):

The paper "Cell-type-specific co-expression inference from single cell RNA-sequencing data" suggests a method to estimate correlation matrix between genes based on the observed UMI count data. The authors consider a Poisson model accounting for variation in sequencing depth, and suggest an IRLS algorithm to estimate the latent covariance matrix. They demonstrate that this algorithm outperforms the competitors by a larger margin.

I found the paper very pleasant to read, as it was very clear and convincing. The IRLS algorithm is simple and statistically sound. Experiments based on simulations and shuffling convincingly show that the author's method outperforms other approaches. I appreciated the use of the IRLS algorithm which I do not often see in the computational scRNAseq literature. I think the paper could be a good fit for Nature Communications.

I list a lot of issues but they should be easily addressable.

MAJOR ISSUES

* I am not very familiar with the literature on estimating correlations in scRNAseq data, so I may very well be unaware of some relevant methods. One relevant method that I do know, is baredSC: <https://bmcbioinformatics.biomedcentral.com/articles/10.1186/s12859-021-04507-8>. It uses Bayesian approach to estimate the bivariate distribution of any two genes, using a very similar Poisson observation model with sequencing depth variation. After the latent bivariate distribution is estimated, the correlation coefficient can be computed. My impression is that it would perform quite well -- perhaps similarly well to CS-CORE, and I think it would be good to include baredSC into the benchmark. I can imagine that it is very slow, so it may not be possible to obtain the entire $p \times p$ correlation matrix for large p .

* If I understood correctly, CS-CORE reports p-values for all off-diagonal entries in the correlation matrix. That is A LOT of p-values: for 10k genes, it's around 50 million significance tests!! It is unclear how the authors deal with the multiple comparison issue (family-wise error? false discovery rate? etc.) I think this should be mentioned already in the section 2.1, perhaps in the summary paragraph of the section.

* The patterns in Figure 1A are very prominent but are left unexplained. Why do almost all methods have average correlation (red lines) above 0? Why are they positive (and not negative)? The only two methods for which it does not hold, are Normalizr and rho-scTransform -- why is that? Can authors provide any insight?

* The CS-CORE line in Figure 2A is "too good"! It looks like a flat line at the true value 0.5. But I am sure that for CS-CORE should also struggle when the expression is too low. This may be an impression because Figure 2A only shows very smoothed lines. Maybe show individual points as well (individual pairs of genes)? Or use less smoothing? Or make the simulation more challenging?

* As can be seen in Figure 2C, rho-scTransform is quite slow. In fact, there is a faster version of scTransform -- see "analytical Pearson residuals" paper <https://genomebiology.biomedcentral.com/articles/10.1186/s13059-021-02451-7>, where the authors argue that it's not only faster but also more sound. This deserves at least a discussion, but ideally a benchmark in Figure 2C.

* I don't understand why Figure 3A uses different ordering of genes in each correlation matrix. I think it may be clearer if all matrices use exactly the same ordering as in the true correlation matrix. Then one could more easily see how well the matrices agree with the true one.

* I don't quite understand why all other methods in Figure 3A work SO BAD. E.g. in rho-scTransform I cannot see any true structure at all. Clearly, if the correlations, the expression strength, and the sequencing depth are all sufficiently high, then all methods should perform okay. Is it because the simulation is very challenging? Can you make it easier and show (perhaps as supplementary figure) that the other methods also work fine then? Currently Figure 3A looks "too good to be true".

* page 22 -- the log-transform formula is given as $\log(x_{ij} / s_j + 1)$, but this transform does not make any sense!! What is usually used, is rather $\log(10000 * x_{ij} / s_j + 1)$, but it is even better to use $\log(\text{median}(s_j) * x_{ij} / s_j + 1)$. See e.g. <https://genomebiology.biomedcentral.com/articles/10.1186/s13059-021-02451-7> but this is discussed in many places. Did you really use $\log(x_{ij} / s_j + 1)$? Then this has to be rerun with a more appropriate log-transformation.

* Related, page 23 -- for Spearman/Pearson correlations, you used "log normalized" data, but how did you do the log transform exactly? Please give the formula. I suggest $\log(\text{median}(s_j) * x_{ij} / s_j + 1)$.

* Large parts of Supplementary Materials look like continuation of the Methods. Why can't all those sections be moved into Methods? Table S1 can also appear in the Methods. And Figure S1 is actually very good, I would suggest to include it into the main paper as Figure 1. It motivates the method really well.

MINOR ISSUES

* Figure 1A: unclear what the x-axis shows. I think this is mentioned in the Methods, but I would suggest to write this in the figure caption.

* page 8: when Pearson and Spearman correlations are mentioned, write explicitly that it's correlation of normalized and log-transformed values.

* Figure 2: some colors are too similar, e.g. the two green colors.

* In Figure 3, did the four groups of genes

* In Section 2.4, emphasize that "cluster" refers to clusters of genes and not clusters of cells! If I understood correctly.

* Figure 3B: how was the clustering done? What clustering algorithm was used? Was this clustering of genes based on the correlation matrix?

* Figure 3: did the simulation include differences in the average expression between the clusters? Or did all genes have the same average expression strength, and the cluster structure was only present in the correlation matrix (which had block-diagonal structure)? This is not sufficiently clear.

* Figure 3: Also, I assume all cells were simulated to be the same, i.e. only one cell type? Please also clarify.

* page 22 -- "we add a regularization step" -- how exactly? Please give more details.

Reviewer #2 (Remarks to the Author):

Su et al. proposes a statistical approach, CS-CORE for estimating and testing co-expression values based on scRNA-seq data. Even though there have been multiple methods for studying the same problem, the proposed method appears to better account for potential biases due to varying sequencing depth or mean expression levels. However, I have multiple concerns about the assumptions and estimation procedures in the statistical method. In addition, the results could be strengthened with more realistic simulation settings and further comparisons.

Methods:

The IRLS procedure is a key step of the proposed CS-CORE algorithm, but is not adequately described. Section S1 gives the objective functions in the regression analysis, but a complete IRLS algorithm for this method should also be provided.

The weight selections for the variance and covariance estimation are not well justified. Even though the manuscript shows that they yield good performance, it is not clear if other choices may lead to better estimation accuracy.

The manuscript explains that the weights for the mean estimation are selected based on statistical efficiency, but what does statistical efficiency refer to in this context?

The proposed method includes a test to assess whether a gene pair have independent expression levels. However, one assumption used to derive the test statistic is that $z_{\{ij\}}$ and $z_{\{ij\}}$ are independent. This assumption does not hold for dependent gene pairs.

Results:

The model uses a Poisson distribution for read counts. How is the performance of the method affected when data follows the negative binomial distribution instead, for example, in Section 4.3?

The manuscript mentions eight methods for comparison, five of which provide statistical tests for co-expression. For those methods not directly providing a test, it would be interesting to see their performance based on a permutation test.

In Figure 3A, the proposed method can reveal the four clusters reflected in the ground truth co-expression, but the estimated values and the true values still demonstrate obvious differences. Therefore, it is surprising that it always estimates the correlation to be 0.5 (which is the same as the true value) in Figure 2A.

The SpQN method also has good performance in results presented in Figure 3, so it would be helpful to add it to the comparisons in Section 2.5 and 2.6.

Review comment for “Cell-type-specific co-expression inference from single cell RNA-sequencing data”

In this work, the authors developed CS-CORE, which is a statistical method for inferring cell-type-specific gene co-expressions. The method builds on a general expression-measurement model that explicitly accounts for sequencing depth variations and measurement errors. The authors demonstrated that CS-CORE can control false positives and can estimate co-expression well, compared with other competing methods. CS-CORE is also computationally fast. The analytical results look convincing and the method is solid. This could be a nice addition to the literature. The followings are my minor comments:

1. In this work, the authors compared CS-CORE with *scran* and many other methods. It would be great if the authors can also compare with methods based on Pearson residuals, including [1].
2. Results from the method locCSN seem to be missing in Figure 2B.

Reference

1. Jan Lause, Philipp Berens & Dmitry Kobak. Analytic Pearson residuals for normalization of single-cell RNA-seq UMI data. *Genome Biology* volume 22, Article number: 258 (2021)

Summary of Revisions

We appreciate all the valuable suggestions from the AE and the reviewers. The manuscript has been substantially revised by adopting all suggestions and addressing all comments. Briefly, we summarize below our main revisions, followed by the point-by-point responses to the reviewers.

- **New methods in benchmarking.** We have incorporated the following new methods in our benchmarking experiments:
 - analytic Pearson residuals (Lause et al., 2021): Please see our response to Reviewer 1 (Point 5) and response to Reviewer 3 (Point 1).
 - baredSC: Please see our response to Reviewer 1 (Point 1).
 - SpQN: Please see our response to Reviewer 2 (Point 8).

CS-CORE continues to yield better performance in the benchmark with the inclusion of these methods.

- **Methods.** As suggested by Reviewers 1 and 2, we have further elaborated on statistical methodology developed in CS-CORE (see response to Reviewer 1, Points 10 and 18; response to Reviewer 2, Points 1-4). In specific, we moved Figure S1, Algorithm S1, Table S1 to the main text to illustrate the confounding effect of sequencing depths, motivate the development of CS-CORE and elaborate on the details of CS-CORE algorithm and real data results, respectively.
- **New experiments in benchmarking.** We have added new simulation experiments to further evaluate statistical power and network clustering accuracy. Please see response to Reviewer 2, Point 6 and response to Reviewer 1, Point 7, respectively.

Response to Reviewer 1

We sincerely thank you for the careful reading of our manuscript and the valuable and constructive comments. In the following, your comments are shown in *italics*, followed by our point-by-point responses. For ease of reference, we have highlighted the major changes in blue in the revised manuscript.

“I found the paper very pleasant to read, as it was very clear and convincing. The IRLS algorithm is simple and statistically sound. Experiments based on simulations and shuffling convincingly show that the author’s method outperforms other approaches. I appreciated the use of the IRLS algorithm which I do not often see in the computational scRNAseq literature. I think the paper could be a good fit for Nature Communications.”

Response: We are grateful for your positive feedback!

Major issues

1. *“I am not very familiar with the literature on estimating correlations in scRNAseq data, so I may very well be unaware of some relevant methods. One relevant method that I do know, is baredSC: <https://bmcbioinformatics.biomedcentral.com/articles/10.1186/s12859-021-04507-8>. It uses Bayesian approach to estimate the bivariate distribution of any two genes, using a very similar Poisson observation model with sequencing depth variation. After the latent bivariate distribution is estimated, the correlation coefficient can be computed. My impression is that it would perform quite well – perhaps similarly well to CS-CORE, and I think it would be good to include baredSC into the benchmark. I can imagine that it is very slow, so it may not be possible to obtain the entire $p \times p$ correlation matrix for large p .”*

Response: Thank you for pointing us to this very relevant reference. Per your suggestion, we have included baredSC into the benchmarking experiment in (i) Figure 1 and Supplementary Fig. 1 that investigated false positive controls using null data with no co-expressions; (ii) Figure 3A and Supplementary Fig. 3,4 that investigated co-expression estimates using simulated data and (iii) Figure 3C that compared computing time. Due to the extreme demand for computing time, baredSC was not

included in experiments with large networks which include the co-expression detection accuracy evaluation in Figure 3B and network clustering accuracy evaluation in Figure 4. In all experiments, baredSC was implemented using the author-provided code at <https://baredsc.readthedocs.io/en/latest/>. We found that baredSC tended to give biased co-expression estimates for both independent and correlated gene pairs.

Specifically, on null data with independent gene expressions, baredSC yielded positive estimates that significantly deviated from zero in both permuted data (Figure 1) and simulated data (Supplementary Fig. 1), even when the sequencing depths were set to be constant across cells. This suggests that the estimation bias in baredSC was due to reasons other than sequencing depth confounding. On simulated data with correlated gene expressions, baredSC had an underestimation bias (Figure 3A). We speculate two possible causes for these biases from baredSC in our experiments. First, the assumption of Gaussian mixtures used to fit log transformed underlying expression levels in baredSC may lead to biased estimates when the underlying distribution cannot be well approximated by a Gaussian mixture model. Second, in baredSC, the MCMC sampling used in inference and the numerical integration used to approximate the likelihood function are both subject to approximation errors.

It is worth noting that baredSC does not provide a statistical test for detecting co-expressions, but it does provide one for determining the signs of co-expressions. Finally, baredSC is indeed slow to compute as you mentioned. In our experiments, we found that it took two minutes to fit baredSC to a single gene pair with only 10 cells (Figure 3C).

We have added baredSC to Figures 2,3A,3C in the revised manuscript, Supplementary Fig. 1,3,4 in the Supplementary Information, and the above discussions to the Results section “CS-CORE has better control of false positive rates” and the Supplementary Notes.

2. *“If I understood correctly, CS-CORE reports p-values for all off-diagonal entries in the correlation matrix. That is A LOT of p-values: for 10k genes, it’s around 50 million significance tests!! It is unclear how the authors deal with the multiple comparison issue (family-wise error? false discovery rate? etc.) I think this should*

Figure R1: An example of CS-CORE p -values before and after BH adjustment to control for false discovery rate. The p -values were evaluated for gene pairs among the top 5,000 highly expressed genes in microglia from control subjects in Lau et al. (2020).

be mentioned already in the section 2.1, perhaps in the summary paragraph of the section.”

Response: Thank you for raising this important point! We have added a sentence at the end of the Results section “Overview of CS-CORE” that stated:

“For all statistical tests performed in in real data analyses, we applied a Benjamini-Hochberg (BH) procedure to control for the false discovery rate.”

While the number of p -values to be evaluated are indeed in the order of p^2 , we found a high percentage of gene pairs remained significant after the BH procedure in our experiments. For example, in Figure 5A, we considered 5,000 genes which resulted in 12,497,500 significance tests for gene pairs in the network. Below is an example of p -values before and after BH adjustment computed by CS-CORE using microglia from control subjects in Lau et al. (2020). We can see a high percentage of gene pairs remained significant after the BH procedure (p -values ≤ 0.05 are shown in the first interval/bin in both plots).

3. “*The patterns in Figure 1A are very prominent but are left unexplained. Why do almost all methods have average correlation (red lines) above 0? Why are they*

Figure R2: Expressions of an independent gene pair in raw UMI counts, scaled and log-normalized counts. We computed the scaled data as $x_i/s_i \times 10^4$ and the log normalized data as $\log(x_i/s_i \times 10^4 + 1)$. This gene pair is representative of the confounding effects by sequencing depth variations in inferring co-expressions with marginally normalized data.

positive (and not negative)? The only two methods for which it does not hold, are Normaliser and rho-scTransform – why is that? Can authors provide any insight?

Response: Thank you for these excellent suggestions. In Figure 1A, the main cause for the overestimation biases from other methods is no or inadequate adjustments of sequencing depth variations when quantifying co-expressions, including the standard log transformations used in locCSN, Pearson correlation of log normalized data, propr and Spearman correlation of log normalized data, and the post-hoc adjustment used in Noise Regularization and SpQN.

To further explain this, we have added a new Figure 2 (previously Supplementary Fig.1) to the Results section “CS-CORE has better control of false positive rates” to illustrate why sequencing depth confounding can lead to overestimation biases. For ease of reference, we include Figure 2 as Figure R2 here.

In Figure R2, we simulated independent UMI counts (details provided in Supplementary Methods) and focused on a pair of genes with relatively high expression levels (these two genes rank 269 and 351 among 28,412 genes in excitatory neurons). We computed the scaled data as $x_i/s_i \times 10^4$ and the log normalized data as $\log(x_i/s_i \times 10^4 + 1)$, and show the scatter plots based on the original, scaled and log normalized counts. It is seen that both scaled data and log normalized data give

spurious positive correlations. The detailed reason is as follows. Given two integers a and b , all cells with UMI counts a, b for these two genes, respectively, are plotted to the point (a, b) in the original UMI counts scatter plot. Interestingly, cells at this point, turning into $(a/s_i \times 10^4, b/s_i \times 10^4)$ after scaling, will be stretched out to form a line with a slope b/a and an intercept 0 in the scaled data, and turning into approximately $(\log(a) - \log(s_i/10^4), \log(b) - \log(s_i/10^4))$ after log normalization, forming a line with a slope 1 and an intercept $\log(b) - \log(a)$ in the log normalized data. These lines are artifacts of the normalization and can seriously inflate false positives when inferring co-expressions.

Normaliser, ρ -analytic Pearson residuals and ρ -sctransform are less subject to such confounding because they apply marginal regressions to explicitly adjust for sequencing depth variations. However, these marginal normalization procedures may not be adequate for co-expression inference, as seen in Figures 3A and Supplementary Fig. 2, where Normaliser, ρ -analytic Pearson residuals and ρ -sctransform had attenuated estimates and reduced power for truly co-expressed gene pairs.

We have added Figure R2 to Figure 2 in the revised manuscript, and the above discussions to the Results section “CS-CORE has better control of false positive rates” in the revised manuscript.

4. *“The CS-CORE line in Figure 2A is “too good”! It looks like a flat line at the true value 0.5. But I am sure that for CS-CORE should also struggle when the expression is too low. This may be an impression because Figure 2A only shows very smoothed lines. Maybe show individual points as well (individual pairs of genes)? Or use less smoothing? Or make the simulation more challenging?”*

Response: Thank you for these insightful questions. CS-CORE in Figure 2A showed a flat line at the true value of 0.5 and this was due to averaging in smoothing. While Figure 2A showed that CS-CORE estimates were unbiased while others were biased, it did not compare the variances of different methods. Indeed as you mentioned, when the expression level was low, CS-CORE estimates, though still averaged to be around 0.5, exhibited larger variances.

In our revision, we added Figure R3 (see next page), that (A) summarized esti-

mates from individual pairs of genes using boxplots across different expression levels, (B) showed the mean squared errors (MSE) of different methods across different expression levels and (C) plotted the average MSE across gene pairs at all expression levels. It can be seen that CS-CORE gave the smallest MSE among all methods.

We have added the above discussions to the Results section “CS-CORE has better co-expression estimation accuracy and detection power” and Supplementary Fig. 4 in the Supplementary Information.

Figure R3: Evaluation of variance and mean squared errors (MSE) of co-expression estimates on gene pairs simulated with a true correlation of 0.5 (5,000 genes and 1,000 cells) under the same setting as Figure 3A. All 11 methods in Figure 3A were evaluated. (A) Boxplots of co-expression estimates against geometric mean expression levels. Gene pairs were stratified by expression levels and boxplots were used to summarize co-expression estimates in the same stratum. Intervals on the x-axis denote the strata of geometric mean expression levels. (B) MSE against geometric mean expression levels. MSE values greater than 0.25 (only found for baredSC) are not plotted. (C) MSE across all gene pairs in the network.

5. “As can be seen in Figure 2C, rho-scTransform is quite slow. In fact, there is a faster version of scTransform – see ”analytical Pearson residuals” paper <https://genomebiology.biomedcentral.com/articles/10.1186/s13059-021-02451-7>, where the authors argue that it’s not only faster but also more sound. This deserves at least a discussion, but ideally a benchmark in Figure 2C.”

Response: Thank you for pointing us to this relevant reference. Per your suggestion and a similar suggestion made by Reviewer 3, we have added analytic Pearson residuals (Lause et al., 2021) into all benchmarking experiments and real data experiments in the Results section. Specifically, we estimated co-expressions using Pearson correlations of analytic Pearson residuals from Lause et al. (2021), referred to as ρ -analytic PR in our work. We found that ρ -analytic PR was faster to calculate and had a similar numerical performance as ρ -scTransform in all experiments.

6. “I don’t understand why Figure 3A uses different ordering of genes in each correlation matrix. I think it may be clearer if all matrices use exactly the same ordering as in the true correlation matrix. Then one could more easily see how well the matrices agree with the true one.”

Response: Thank you for your suggestion. We chose to reorder the genes by hierarchical clustering results as we also wanted to show the clustering performance from different methods. We agree with you that this can make it difficult to compare the estimated co-expressions with the ground truth. Per your suggestion, we have added Figure R4 (see next page) which fixed the gene ordering to be the same as that for the true correlation matrix for all methods. It is seen that CS-CORE is the closest to the truth. In our revision, we added this Figure as Supplementary Fig. 5. Figure 4A and this new Supplementary Fig. 5 together give a complete comparison between CS-CORE and other methods in both network estimation and clustering accuracy.

7. “I don’t quite understand why all other methods in Figure 3A work SO BAD. E.g. in rho-scTransform I cannot see any true structure at all. Clearly, if the correlations, the expression strength, and the sequencing depth are all sufficiently high, then all methods should perform okay. Is it because the simulation is very challenging?”

Figure R4: Heatmaps of true and estimated co-expression networks from simulations under the same setting as Figure 4A. For all methods, genes were ordered the same as the true co-expression network. CS-CORE is compared to locCSN, Noise Regularization, Normalizr, Pearson correlation of log normalized data (Pearson), propr, Spearman correlation of log normalized data (Spearman), SpQN, ρ -analytic Pearson residuals (ρ -analytic PR) and ρ -sctransform.

Can you make it easier and show (perhaps as supplementary figure) that the other methods also work fine then? Currently Figure 3A looks "too good to be true".

Response: Thank you for your questions and suggestions. The other methods in Figure 3A did not work well as they are subject to the confounding effects by sequencing depth variations and the attenuation bias due to measurement errors, as described in the Results section in the main text. In what follows, we add more explanations on the simulation setting in Figure 3A, which is more in line with real single cell data, and two new simulation experiments with less challenging parameter settings, where the performance of other methods improved.

First, the setting in Figure 3A considered cells with varying sequencing depths (min=407, max=19,998, median=5,318) and 100 genes randomly sampled from the top 2,000 highly expressed genes in real data. The sequencing depths and expression levels in this experiment are in fact typical in real data scenarios. However, under this setting, most existing methods tended to generate inflated co-expression estimates on independent gene pairs, as seen in Figures 1 and 2, and attenuated estimates on co-expressed gene pair, as seen in Figure 3. These two sources of bias together prevented existing methods from accurately recovering true co-expressed clusters.

To further demonstrate this, we added two new sets of simulation experiments where we focused on genes with extremely high expression levels, such that the attention bias was much reduced, and let the sequencing depth to be constant across cells, such that the sequencing depth confounding no longer existed. Specifically, in the first experiment, we considered genes with extremely high expression levels such that estimates from existing methods were much less attenuated, as demonstrated in Figure 3A. Under this setting, it is seen from Figure R5A (see next page) that estimates from most existing methods improved. However, locCSN and propr still performed poorly due to sequencing depth confounding. In the second experiment, we further set the sequencing depths to be constant across cells. As shown in Figure R5B (see next page), the estimates of locCSN and propr improved and all existing methods gave clustering results that are close to the truth. In our revision, we have added these new figures to Supplementary Fig.6 and discussions to the Results section “Other methods give biased results in downstream co-expression analysis”.

8. “Page 22 – the log-transform formula is given as $\log(x_{ij}/s_j + 1)$, but this transform does not make any sense!! What is usually used, is rather $\log(10000 \cdot x_{ij}/s_j + 1)$, but it is even better to use $\log(\text{median}(s_j) * x_{ij}/s_j + 1)$. See e.g. <https://genomebiology.biomedcentral.com/articles/10.1186/s13059-021-02451-7> but this is discussed in many places. Did you really use $\log(x_{ij}/s_j + 1)$? Then this has to be rerun with a more appropriate log-transformation.”

Response: Thank you for your careful reading. There was a typo in the formula in our original manuscript. We have been using $\log(10^4 \times x_{ij}/s_j + 1)$ to calculate log normalized gene expressions in all data analyses. This typo is now fixed.

9. “Related, page 23 – for Spearman/Pearson correlations, you used “log normalized” data, but how did you do the log transform exactly? Please give the formula. I suggest $\log(\text{median}(s_j) * x_{ij}/s_j + 1)$.”

Response: Thank you for your suggestion. We have added the formula for calculating log normalized data, i.e., $\log(10^4 \times x_{ij}/s_j + 1)$ in the Methods section when describing the Pearson and Spearman correlations. Multiplying the scaled counts with 10000 is another common choice, adopted in popular pipelines such as Seurat.

Figure R5: Evaluation of CS-CORE in recovering co-expressed gene clusters using simulated data with (A) high expression levels for all genes and varying sequencing depths across cells; (B) high expression levels for all genes and constant sequencing depths across cells, compared to locCSN, Noise Regularization, Normalisr, Pearson correlation of log normalized data (Pearson), propr, Spearman correlation of log normalized data (Spearman), SpQN, ρ -analytic PR and ρ -sctransform. Heatmaps of true and estimated co-expression networks are shown, where genes were ordered by applying hierarchical clustering to the estimated co-expression network and color coded by their true gene cluster labels. The high expression levels were set by the top 100 highly expressed genes in excitatory neurons from Lau et al. (2020). The constant sequencing depth was set to 5318.

10. *“Large parts of Supplementary Materials look like continuation of the Methods. Why can’t all those sections be moved into Methods? Table S1 can also appear in the Methods. And Figure S1 is actually very good, I would suggest to include it into the main paper as Figure 1. It motivates the method really well.”*

Response: Thank you for these suggestions! In our revision, we have moved Figure S1 and related discussions to Figure 2 and the Results section “CS-CORE has better control of false positive rates” in the main text to illustrate that sequencing depth variations cannot be adjusted by standard marginal normalization. We have also moved Table S1 and Algorithm S1 to the Methods section in the revised manuscript.

Minor issues

11. *“Figure 1A: unclear what the x-axis shows. I think this is mentioned in the Methods, but I would suggest to write this in the figure caption.”*

Response: Thank you for your suggestion. We have updated the caption of Figure 1A to add that the x-axis shows the mean expression levels of genes.

12. *“Page 8: when Pearson and Spearman correlations are mentioned, write explicitly that it’s correlation of normalized and log-transformed values.”*

Response: Thank you for your suggestion. We have added “of log normalized data” at all places that Pearson correlations and Spearman correlations were mentioned.

13. *“Figure 2: some colors are too similar, e.g. the two green colors.”*

Response: We have updated the color panels in all relevant figures. With the new color panel, CS-CORE is the only method colored by green.

14. *“In Section 2.4, emphasize that “cluster” refers to clusters of genes and not clusters of cells! If I understood correctly”*

Response: You are right and they are indeed clusters of genes. We have updated the Results section “Other methods give biased results in downstream co-expression

analysis” and Figure 4 to clarify that the clusters refer to clusters of co-expressed genes.

15. *“Figure 3B: how was the clustering done? What clustering algorithm was used? Was this clustering of genes based on the correlation matrix?”*

Response: Thank you for your questions. The clustering in Figure 3 (now Figure 4) was performed by applying hierarchical clustering on the estimated correlation matrix with the number of clusters set to 4. We have added these details to the caption of Figure 4 in our revision.

16. *“Figure 3: did the simulation include differences in the average expression between the clusters? Or did all genes have the same average expression strength, and the cluster structure was only present in the correlation matrix (which had block-diagonal structure)? This is not sufficiently clear.”*

Response: Thank you for your careful reading. For the experiment in Figure 3 (now Figure 4), the mean expression levels of genes were randomly sampled from the top 2,000 highly expressed genes in real data. Correspondingly, the cluster structure was only present in the correlation matrix and gene expression levels were similar across gene clusters, with mean expression levels (on the scale of $\log_{10} \mu_j + 3$) -0.55 with an sd=0.36 for cluster 1, -0.55 with an sd=0.45 for cluster 2, -0.57 with an sd=0.39 for cluster 3 and -0.70 with an sd=0.26 for cluster 4. We have added the above clarifications to the first section of Supplementary Methods.

17. *“Figure 3: Also, I assume all cells were simulated to be the same, i.e. only one cell type? Please also clarify.”*

Response: You are right and all cells were simulated from the same cell type. We have added a sentence in the Methods section “Simulating from the multivariate expression-measurement model” to clarify this.

18. *“Page 22 – “we add a regularization step” – how exactly? Please give more details. ”*

Response: Thank you for your question. Calculating the weights for the weighted

least squares in CS-CORE requires the estimated variance $\hat{\sigma}_{jj}$ for $j = 1, \dots, p$. See the Methods section in the manuscript for detailed formulas of the weights. For instance, we derived that the optimal weight, in terms of minimizing the variance, when estimating the mean parameter is $w_{ij} = 1/(s_i\mu_j + s_i^2\sigma_{jj})$. In practice, we find that estimated $\hat{\sigma}_{jj}$'s can be variable, leading to unstable weight estimates and an increased variance of the estimator. To address this issue, we write $\sigma_{jj} = \mu_j^2 \times \theta_j$, where $\theta_j = \sigma_{jj}/\mu_j^2$ is the over-dispersion parameter to be regularized across genes. A similar idea has been considered in DESeq2 (Love et al., 2014) and sctransform (Hafemeister and Satija, 2019). Specifically, we set $\hat{\theta}_j^{(t)} = \text{median}_l\{(\hat{\sigma}_{jj})^{(t-1)}/(\hat{\mu}_j^{(t-1)})^2\}$ at iteration t , that is, all genes share the same over-dispersion parameter at each iteration of the iteratively reweighted least squares. We found that such a simple regularization step leads to stable weight estimates and a reduced variance of the weighted least squares estimator.

In our revision, we have moved Algorithm S1 from the Supplementary Information to the main text (now Algorithm 1), where line #12 details how regularization is applied when estimating the weights in CS-CORE. We have also added the above discussions to the Methods section “CS-CORE method” in the revised manuscript.

Response to Reviewer 2

We greatly appreciate your valuable suggestions. In the following, your comments are shown in *italics*, followed by our point-by-point responses. For ease of reference, we have highlighted the major changes in blue in the revised manuscript.

1. *“The IRLS procedure is a key step of the proposed CS-CORE algorithm, but is not adequately described. Section S1 gives the objective functions in the regression analysis, but a complete IRLS algorithm for this method should also be provided.”*

Response: Thank you for this suggestion. We have moved Algorithm S1 from the Supplementary Information to the main text (now Algorithm 1). It gives the complete IRLS algorithm used in CS-CORE for parameter estimation.

2. *“The weight selections for the variance and covariance estimation are not well justified. Even though the manuscript shows that they yield good performance, it is not clear if other choices may lead to better estimation accuracy.”*

Response: Thank you for your insightful comments. Under our framework, optimal weights, in terms of minimizing variances, for the variance and covariance weighted least squares estimators are difficult to derive analytically as they require fourth moment quantities such as $\text{Var}(x_{ij}^2)$ or $\text{Var}(x_{ij}x_{ij'})$, where x_{ij} is the UMI counts from gene j in sample i . These higher moments need to be derived under explicit parametric assumptions on the underlying expression levels. In CS-CORE, we did not place such assumptions as they can limit the flexibility of our approach. Instead, we set weights for the variance and covariance estimators to be $\{\text{Var}(x_{ij})\}^{-2}$ and $\{\text{Var}(x_{ij})\text{Var}(x_{ij'})\}^{-1}$ based on the rationale that, when estimating σ_{jj} or $\sigma_{jj'}$, the sample with a larger variance in counts should have a lower weight. Indeed, we found that such weight parameters performed well in experiments and were fast to compute.

You made an excellent point on trying out other weight choices. We had in fact experimented with other weights when developing CS-CORE. For example, we derived the optimal weights for the variance and covariance estimators under a negative binomial assumption on x_{ij} 's. In this case, the optimal weight of, for example, the variance estimator is $\{\text{Var}(x_{ij}) + (2 + 6\sigma_{jj}/\mu_j^2) \cdot \text{Var}(x_{ij})^2\}^{-1}$. We can see this is sim-

Figure R6: Comparison of mean μ_j and variance σ_{jj} estimated by iteratively reweighted least squares using the CS-CORE weights and the weights derived under the negative binomial (NB) assumption. The expression data were obtained as described in Figure 1.

ilar to our selected weight of $\{\text{Var}(x_{ij})\}^{-2}$ if σ_{jj}/μ_j^2 's are similar across genes. In our experiments, we found that the derived weights under the negative binomial assumption are slower to compute due to their involved formulas and yielded very similar numerical performances, as shown in Figure R6 above. Hence, we had chosen to use the current weights for variance and covariance estimators in CS-CORE.

We have added the above discussions to Supplementary Notes.

3. “*The manuscript explains that the weights for the mean estimation are selected based on statistical efficiency, but what does statistical efficiency refer to in this context?*”

Response: Thank you for the question. The mean parameter μ_j is related to the observed data through $x_{ij} = s_i\mu_j + \epsilon_{ij}$, where $\mathbb{E}(\epsilon_{ij}) = 0$, and is estimated via the following weighted least squares

$$\min_{\mu_j} \sum_i w_{ij} (x_{ij} - s_i\mu_j)^2.$$

We set the weights $w_{ij} = 1/\text{Var}(\epsilon_{ij})$ where ϵ_{ij} is defined above. This choice yields a statistically efficient estimator in the sense that the weighted least squares esti-

mator for μ_j with weight w_{ij} is the estimator with the *smallest variance* among all linear unbiased estimators (Gauss–Markov theorem; Aitken (1936)). We have added a sentence to the Methods section “CS-CORE method” to clarify this point.

4. “*The proposed method includes a test to assess whether a gene pair have independent expression levels. However, one assumption used to derive the test statistic is that z_{ij} and $z_{ij'}$ are independent. This assumption does not hold for dependent gene pairs.*”

Response: Thank you for raising this point. To clarify, independent z_{ij} and $z_{ij'}$ is not an assumption in our testing procedure but the null hypothesis. Specifically, we aim to test for, j, j' ,

H_0 : z_{ij} and $z_{ij'}$ are independent,

H_a : z_{ij} and $z_{ij'}$ are dependent.

Under the null hypothesis, we derive the distribution of the test statistic $T_{jj'}$, which is shown to asymptotically follow a standard normal distribution. Hence, when z_{ij} and $z_{ij'}$ are independent (under H_0), rejecting H_0 if $|T_{jj'}| > \Phi_{0.975}$, where Φ_α denotes the α -th quantile of a standard normal distribution, gives the appropriate type I error control at 0.05. That is, the probability of rejecting the null hypothesis given that it is true is 0.05.

When z_{ij} and $z_{ij'}$ are dependent (under H_a), the test statistic is expected to deviate from the null distribution, leading to rejection of the H_0 . This is characterized as the power of the test. Under our setting, it is difficult to analytically derive the power as our approach does not make parametric assumptions on the distribution of z_{ij} 's, so it is difficult to characterize the distribution of the test statistic under the alternative hypothesis. Empirically, in our numerical experiments with varying settings, we found that the test in CS-CORE has more power than the tests offered in other methods (Supplementary Fig. 2) while having an appropriate type-I error control (Figure 1 and Supplementary Fig.1). Theoretically analyzing the power of our proposed test is an important task and we plan to investigate this in our future research.

5. “*The model uses a Poisson distribution for read counts. How is the performance*

of the method affected when data follows the negative binomial distribution instead, for example, in Section 4.3?"

Response: Thank you for your question. We wish to clarify that, while our model includes a Poisson distribution, the UMI counts under our model are usually not Poisson.

Recall that we assume the following model for UMI counts:

$$x_{ij}|z_{ij} \sim \text{Poisson}(s_i z_{ij}), \quad (z_{i1}, \dots, z_{ip}) \sim F_p(\cdot), \quad (\text{R1})$$

where x_{ij} is the UMI count of gene j in cell i , z_{ij} is the underlying expression level, assumed to follow an unknown nonnegative p -variate distribution $F_p(\cdot)$, and s_i is the sequencing depth. As the underlying expressions z_{ij} 's are random variables, the distribution of x_{ij} is usually not Poisson and can have a high dispersion level depending on $F_p(\cdot)$. In fact, if $F_p(\cdot)$ is Gamma, the distribution of count x_{ij} is negative binomial under (R1) (Sarkar and Stephens, 2021), which is the distribution you suggested when modeling UMI counts.

In CS-CORE, we do not place explicit parametric assumptions on $F_p(\cdot)$, and use a moment based method to estimate co-expressions $\text{Cov}(z_{ij}, z_{ij'})$'s for $1 \leq j, j' \leq p$. Hence, our model includes modeling x_{ij} 's using negative binomial as a special case and is more flexible.

In Equation (R1), we choose Poisson to model the conditional distribution of $x_{ij}|z_{ij}$, referred to as the measurement model in CS-CORE, as it agrees with the existing literature (Hafemeister and Satija, 2019; Lause et al., 2021; Sarkar and Stephens, 2021) that a Poisson distribution is usually sufficient to characterize the variations introduced by the sequencing experiment, modeled as $x_{ij}|z_{ij}$ in our approach. In the case that mRNA molecules are sequenced with more variations than Poisson, it can be useful to adapt CS-CORE to model $x_{ij}|z_{ij}$ using other distributions that model nonnegative integers such as the negative binomial distribution. This requires re-deriving the moment conditions used in CS-CORE, which is given in Equation (2) in the paper, under the specific distribution used to model $x_{ij}|z_{ij}$, and updating the estimating and testing results accordingly. This is an important direction that requires a thorough and separate investigation, and we leave it to future work. We have added

Figure R7: Statistical power in detecting co-expression at different gene expression levels. For the three methods with appropriate type-I error controls in Figure 1B (CS-CORE, Normalizr and ρ -sctransform), the analytical power was evaluated based on the proposed statistical tests and plotted with solid lines. For methods with inflated type-I errors (Noise regularization, Pearson correlations of log normalized data, Spearman correlations of log normalized data and ρ -analytic PR) or methods that do not offer statistical tests (propr, SpQN), the empirical power was evaluated based on empirical p -values as described in Supplementary Methods and plotted with dashed lines. Gene expressions were simulated under the same setting as in Figure 3A. The lines show the average power across expression levels fitted by kernel smoothing.

the above discussions to the Discussion section of the revised manuscript.

6. “The manuscript mentions eight methods for comparison, five of which provide statistical tests for co-expression. For those methods not directly providing a test, it would be interesting to see their performance based on a permutation test.”

Response: Thank you for your excellent suggestion, which helped to strengthen the power comparison between CS-CORE and other methods. In our revision, we have added methods that do not directly provide a test to our power comparison, by doing a computation-based tests, as shown in Figure R7 (also Supplementary Fig.2). locCSN and baredSC were not included in the comparison due to their extreme computational demands. It can be seen that the power of CS-CORE is the highest across varying expression levels.

Next, we explain why a permutation test cannot be easily implemented under our

Figure R8: True mean μ_j and variance σ_{jj} parameters versus their estimates from permuted and Poisson sampled data. Gene expression data were simulated as in Figure 3A and the permuted and Poisson sampled data were generated following the Methods section.

setting and add some details on the computation-based test we used. Specifically, the standard permutation test can be carried out in three steps: (1) for each gene, permute its expressions across cells, such that gene expressions are decorrelated; (2) estimate co-expressions using a specific method on these permuted data to find the null distribution; (3) compare the co-expression estimates calculated from the observed data to the null distribution to find the empirical p -value. For scRNA-seq data, step (1) is problematic as different cells have different sequencing depths and gene counts are not exchangeable across different cells. In the Results section “CS-CORE has better control of false positive rates”, we used an approach that combined permutation with Poisson sampling, where we permuted normalized expressions, to generate null data where gene pairs are not co-expressed. However, we find that this approach is not appropriate for calculating the null distribution in a test, due to the over-dispersion caused by the Poisson sampling step (Figure R8). Alternatively, we consider a simulation-based approach that simulates independent gene expressions with mean expression levels and variances that resemble real data (see Supplementary Methods) and estimates the empirical null distribution using co-expressions estimated from the simulated null data.

We have added the above discussion and results to the Results section “CS-

CORE has better control of false positive rates” and the Methods section “Other co-expression estimation and testing methods using scRNA-seq data” in the revised manuscript and Supplementary Methods in Supplementary Information. We have also added Figure R7 to Supplementary Fig. S2.

7. *“In Figure 3A, the proposed method can reveal the four clusters reflected in the ground truth co-expression, but the estimated values and the true values still demonstrate obvious differences. Therefore, it is surprising that it always estimates the correlation to be 0.5 (which is the same as the true value) in Figure 2A.”*

Response: Thank you for your careful reading. The CS-CORE curve in Figure 2A is smoothed over estimates from all gene pairs with similar expression levels, while the estimate from a specific gene pair varies around the true correlation of 0.5 (see Figure R3 on page 8 of this response letter). In Figure 3A (now Figure 4A), the difference between CS-CORE estimates and the ground truth reflects this variation. The variation of CS-CORE estimates around the true value is expected to decrease as expression level increases (Figure R3) and as sample size increases.

We have added the above clarification and result to the Results section “CS-CORE has better co-expression estimation accuracy and detection power” and Supplementary Fig. 4.

8. *“The SpQN method also has good performance in results presented in Figure 3, so it would be helpful to add it to the comparisons in Section 2.5 and 2.6.”*

Response: Thank you for your suggestion. In our revision, we have added SpQN to the clustering analysis comparisons in the Results sections “CS-CORE identified more biologically relevant co-expressions from AD samples” and “CS-CORE identified upregulated co-expressions from COVID-19 blood samples”. Specifically, we extracted (differentially) co-expressed modules from SpQN estimates, applied GO enrichment analyses and compared the GO functions enriched in SpQN modules with those from CS-CORE. We found that SpQN in general yielded fewer modules enriched for relevant cell-type-specific biological functions across all cell types. These results are detailed in Supplementary Data 1-4. We also evaluated SpQN estimates on genes from four GO terms on microglia’s function as in Figure 5C and found

that SpQN could only partially recover gene modules characterized by these four GO terms, while CS-CORE could accurately group all genes into respective GO terms, and the estimated co-expressions from SpQN were weaker than those from CS-CORE (Supplementary Fig. 10).

Response to Reviewer 3

We appreciate your valuable suggestions. In the following, your comments are shown in *italics*, followed by our point-by-point responses. For ease of reference, we have highlighted the major changes in blue in the revised manuscript.

“The authors demonstrated that CS-CORE can control false positives and can estimate coexpression well, compared with other competing methods. CS-CORE is also computationally fast. The analytical results look convincing and the method is solid. This could be a nice addition to the literature.”

Response: We really appreciate your positive review.

1. *“In this work, the authors compared CS-CORE with sctransform and many other methods. It would be great if the authors can also compare with methods based on Pearson residuals, including Lause et al. (2021).”*

Response: Thank you for pointing us to this relevant reference. Per your suggestion and a similar suggestion made by Reviewer 1, we have added analytic Pearson residuals (Lause et al., 2021) into all benchmarking experiments and real data experiments in the Results section. Specifically, we estimate co-expressions using Pearson correlations of analytic Pearson residuals from Lause et al. (2021), referred to as ρ -analytic PR in our work. We found that ρ -analytic PR was faster to calculate and had a similar numerical performance as ρ -sctransform in all experiments.

2. *“Results from the method locCSN seem to be missing in Figure 2B.”*

Response: Thank you for your careful reading. The method locCSN was missing from Figure 2B due to its extreme demand for computing time. For example, it took several minutes to fit locCSN to a single gene pair. As a result, it is difficult to evaluate locCSN under the setting of Figure 2B (now Figure 3B), where a co-expression network with 5,000 genes (12,497,500 pairs) needs to be estimated in 100 data replicates.

References

- Aitken, A. C. (1936), “IV.—On least squares and linear combination of observations,” *Proceedings of the Royal Society of Edinburgh*, 55, 42–48.
- Hafemeister, C. and Satija, R. (2019), “Normalization and variance stabilization of single-cell RNA-seq data using regularized negative binomial regression,” *Genome biology*, 20, 1–15.
- Lau, S.-F., Cao, H., Fu, A. K., and Ip, N. Y. (2020), “Single-nucleus transcriptome analysis reveals dysregulation of angiogenic endothelial cells and neuroprotective glia in Alzheimer’s disease,” *Proceedings of the National Academy of Sciences*, 117, 25800–25809.
- Lause, J., Berens, P., and Kobak, D. (2021), “Analytic Pearson residuals for normalization of single-cell RNA-seq UMI data,” *Genome biology*, 22, 1–20.
- Love, M. I., Huber, W., and Anders, S. (2014), “Moderated estimation of fold change and dispersion for RNA-seq data with DESeq2,” *Genome biology*, 15, 1–21.
- Sarkar, A. and Stephens, M. (2021), “Separating measurement and expression models clarifies confusion in single-cell RNA sequencing analysis,” *Nature genetics*, 53, 770–777.

REVIEWERS' COMMENTS

Reviewer #1 (Remarks to the Author):

I thank the authors for very carefully addressing all issues that were raised by the reviewers. I think the paper could be published as is.

A tiny nitpick is that in Supp Fig 4B/C, locCSN is colored inconsistently (different colors in B and in C).

One further suggestion would be to discuss Sanity (<https://www.nature.com/articles/s41587-021-00875-x>) -- a Bayesian method to estimate latent gene expression that is advertised to yield good estimates of correlations as well (see Fig 4 in the Sanity paper). It could be interesting to compare CS-CORE with Pearson correlation on Sanity estimates. However, as this did not come in the first round of the revision, I don't insist on this additional analysis.

Reviewer #2 (Remarks to the Author):

The revised manuscript has addressed all my questions.

Reviewer #3 (Remarks to the Author):

All my previous comments have been reasonably addressed. I do not have any further comments.

Response to Reviewer 1

“I thank the authors for very carefully addressing all issues that were raised by the reviewers. I think the paper could be published as is.

A tiny nitpick is that in Supp Fig 4B/C, locCSN is colored inconsistently (different colors in B and in C).

One further suggestion would be to discuss Sanity – a Bayesian method to estimate latent gene expression that is advertised to yield good estimates of correlations as well (see Fig 4 in the Sanity paper). It could be interesting to compare CS-CORE with Pearson correlation on Sanity estimates. However, as this did not come in the first round of the revision, I don't insist on this additional analysis.

Response: Thank you for your careful reading! We have updated the color panels in Supplementary Figures 4B and C to use consistent colors for locCSN. We also thank you for introducing this Sanity method. We agree that it would be interesting to compare co-expressions estimated by Sanity with our method CS-CORE, and we will investigate this in our future work.

Response to Reviewer 2

“The revised manuscript has addressed all my questions. ”

Response: Thank you for your positive feedback!

Response to Reviewer 3

“All my previous comments have been reasonably addressed. I do not have any further comments. ”

Response: Thank you for your positive feedback!